# Synergistic antibacterial action of AgNP-ampicillin conjugates: Evading β-lactamase degradation in ampicillin-resistant clinical isolates

**Muhammad Salehuddin Ayubee** [1,2], **Farhana Akter**[1,3], **Nadia Tasnim Ahmed**[1],
**Abul Kalam Lutful Kabir**[4], **Md. Mahboob Hossain**[5], **Muhammad Delwar Hussain**[6],
**Mohsin Kazi**[7], **Md. Abdul Mazid**[1]*

1 Department of Pharmaceutical Chemistry, Faculty of Pharmacy, University of Dhaka, Dhaka,
Bangladesh, 2 Department of Pharmacy, Northern University Bangladesh, Dhaka, Bangladesh,
3 Department of Pharmacy, East West University, Dhaka, Bangladesh, 4 Department of Pharmaceutical
Technology, Faculty of Pharmacy, University of Dhaka, Dhaka, Bangladesh, 5 Microbiology Program,
Department of Mathematics and Natural Sciences, BRAC University, Dhaka, Bangladesh, 6 Department
of Pharmaceutical Sciences, School of Pharmacy, University of Maryland Eastern Shore, Maryland, United
States of America, 7 Department of Pharmaceutics, College of Pharmacy, King Saud University, Riyadh,
Saudi Arabia,

* ma.mazid@du.ac.bd

doi.org/10.1371/journal.pone.0331669

of Science and Technology, VIETNAM

**Peer Review History:** PLOS recognizes the
benefits of transparency in the peer review
process; therefore, we enable the publication
of all of the content of peer review and
author responses alongside final, published
articles. The editorial history of this article is
available here: https://doi.org/10.1371/journal.
pone.0331669

## Abstract

### Objectives

Antibiotic resistance towards penicillin has been attempted to counter by chemically
modifying ampicillin through the conjugation with silver nanoparticles (AgNPs). The
current study optimizes the conditions for synthesizing and characterizing AgNP-
ampicillin to quantify the conjugation extent, evaluate the antibacterial efficacy, and
explore the underlying antibacterial mechanisms.

### Materials and methods

AgNPs were synthesized from silver nitrate by chemical reduction method,
silica-coated with tetraethyl orthosilicate (TEOS) and amine functionalized by
(3-aminopropyl) triethoxysilane (APTES), which was then conjugated with ampicillin
via the carbodiimide chemistry. UV-visible spectroscopy and DLS were employed to
confirm the synthesis of AgNPs, while FT-IR and TGA were used to confirm ampicillin
functionalization and conjugation, and SEM and EDX spectroscopy provided morpho-
logical insight. Microbial assays were conducted against *Bacillus subtilis*, *Escherichia
coli*, *Staphylococcus aureus*, and *Pseudomonas aeruginosa* to determine the inhibi-
tion zones, MIC, MBC, MPC, MBIC, MBEC, FIC index, and time-dependent efficacy
of AgNP-ampicillin. Cytotoxicity was assessed on Vero cells while molecular docking
was performed using AutoDock Vina.

**Data availability statement:** All relevant data are within the paper and its Supporting Information files.

**Funding:** This study was supported by the Grants for Advanced Research in Education (GARE) program (No. 37.20.0000.004.033.020.2016.7725) of Ministry of Education, Government of the People's Republic of Bangladesh. The funders had no role in study design, data collection and analysis, decision to publish, or preparation of the manuscript.

**Competing interests:** The authors have declared that no competing interests exist.

**Abbreviations:** DLS, Dynamic light scattering, FT-IR, Fourier transform infrared spectroscopy, TGA, Thermogravimetric analysis, SEM, Scanning electron microscope (SEM), EDX, Energy dispersive X-ray, MIC, Minimum inhibitory concentration, MBC, Minimum bactericidal concentration, MPC, Mutant prevention concentration, MBIC, Minimum biofilm inhibitory concentration, MBEC, Minimum biofilm eradication concentration, FIC, Fractional inhibitory concentration.

## Results and discussions

The synthesized conjugates demonstrated an approximate conjugation efficiency of 57.7%, with four ampicillin molecules conjugated per AgNP. The AgNP-ampicillin conjugates exhibited enhanced antibacterial activity against the studied clinical isolates compared to AgNPs or ampicillin alone, as evidenced by significant differences in inhibition areas in One-way ANOVA (F=18-25.68, p<0.05), while Tukey's post-hoc analysis suggested synergistic effects. AgNP-ampicillin demonstrated enhanced bacteriostatic and bactericidal activity against both planktonic and biofilm-forming cells with mutant prevention ability, and upto 1.25 times faster bacterial elimination compared to ampicillin and AgNPs alone. Synergistic effects of AgNP-ampicillin were confirmed by an FIC index (≤0.5), and effective protection of ampicillin from β-lactamase degradation was established through molecular docking. Cytotoxicity testing confirmed >95% Vero cell viability, indicating minimal toxicity.

## Conclusion

The AgNP-ampicillin conjugates exhibited enhanced antibacterial efficacy, biofilm disruption, and protection against β-lactamase degradation while maintaining low toxicity.

## 1. Introduction

The continuous rise in antibiotic resistance has undermined the efficacy of many clinically used drugs, including β-lactam antibiotics. More than two out of five strains of *Staphylococcus aureus (S. aureus)* sampled from clinics are methicillin-resistant *S. aureus* (MRSA), including a few strains that were also alarmingly resistant to vancomycin [1,2]. This resistance is mainly mediated by the production of β-lactamase enzymes that hydrolyze the β-lactam ring, making the antibiotic ineffective [1,3]. It is thought that MRSA infections preferentially establish themselves in biofilms by creating a protective extracellular matrix and inducing a dormant state in bacterial cells, which diminishes the efficacy of antibiotics and host immune system that target actively dividing cells [2,4]. These characteristics established MRSA as one of the most common causes of nosocomial infection that increases morbidity, mortality, and costs to the health system [1].

Like other β-lactam antibiotics, ampicillin binds with the DD-transpeptidase enzyme, inhibiting the peptidoglycan cross-linking in the bacterial cell wall, leading to bacterial cell death by increasing osmotic pressure [5]. However, this process for bacterial death triggers the activation of β-lactamase enzyme in the bacteria, leading to antibiotic resistance [5]. Bacterial intrinsic resistance to ampicillin is also a significant concern in antimicrobial therapy, since many Enterobacteriaceae, such as *Enterobacter* spp. and *Serratia* spp. possess chromosomally encoded AmpC beta-lactamases that confer intrinsic resistance to ampicillin and other beta-lactam antibiotics [6]. In *Enterococcus faecium*, high-level ampicillin resistance is primarily

attributed to enhanced production or mutations of PBP5 [7]. The emergence of bacterial resistance has significantly influenced empiric therapy choices, leading to the use of combination therapies or newer antibiotics [8].

The limited efficacy of vancomycin, currently used as the antibiotic of last resort for MRSA infections, is further compromised by the emergence of vancomycin-resistant strains (VRSA). It makes infections more challenging to treat, often leading to higher mortality rates and limited antibiotic options in already vulnerable patients, highlighting an urgent need for alternative treatment options [1,2]. While developing alternative antibiotics or chemical modifications of existing drugs is explored, they remain prone to rapid bacterial resistance, limiting their clinical utility [9–13]. Therefore, there is an urgent need for novel antibacterial strategies that promote the efficiency of presently existing antibiotics [12,13]. Given the challenges in developing new antibiotics, innovative approaches, such as nanotechnology, offer a promising solution [14].

Metallic nanoparticles (MNPs) are known to have antimicrobial properties and can be utilized in controlling infectious diseases [15]. Nanoparticles serve both as antimicrobial and a drug delivering system due to their small size. MNPs can penetrate bacterial cell walls and biofilms effectively, and the inherent bactericidal activity through generation of reactive oxygen species (ROS) works complementary to the antibacterial effect of antibiotics [16]. These properties make nanoparticles a promising solution to address resistance mechanisms, particularly when conjugated with existing antibiotics to maximize synergistic effects [17]. Some researchers describe AgNPs as the "new generation of antimicrobials", as the bactericidal efficacy of AgNPs was explored against *S. aureus* as well as *Escherichia coli (E. coli),* where AgNPs interact with the bacterial cell membrane [3,18–20]. Meanwhile, silver sulfadiazine has been replaced by nanosilver sulfadiazine as an active agent for treating wounds, and efforts have been made to incorporate AgNPs into various medical and surgical equipment including protective masks [21].

Despite their promising antibacterial properties, the standalone use of AgNPs are associated with significant challenges, including cytotoxicity to mammalian cells, potential environmental toxicity, and a lack of targeted action [22–24]. This leads to the development of conjugates combining AgNPs with conventional antibiotics to enhance their efficacy. Furthermore, conjugating nanoparticles with antibiotics, such as ampicillin, can enhance antibacterial efficacy through synergistic effects, improved pharmacokinetics, and resistance mitigation [25,26]. Studies have demonstrated that the antimicrobial activity of ampicillin was boosted against resistant strains of *S. aureus, Pseudomonas aeruginosa (P. aeruginosa)*, and *Klebsiella aerogenes* (*K. aerogenes*) when directly conjugated with silver or gold nanoparticles [27,28]. In another study, amine group-functionalized silver nanoparticles conjugated with carboxylic group-functionalized ampicillin showed increased bactericidal activity [29].

Recent studies have investigated the conjugation of AgNPs with ampicillin against multidrug-resistant bacteria. The ampicillin-AgNPs exhibit a better antibacterial potency against Gram-positive and Gram-negative bacteria than the antibiotic or AgNPs alone [30]. Likewise, amoxicillin-conjugated AgNPs were the best against *Streptococcus pyogenes* and *Haemophilus influenzae* and prevented biofilm formation in endotracheal tubes [31]. Overall, ampicillin-AgNPs exhibited lower minimum inhibitory concentrations than ampicillin or chemically synthesized AgNPs, and no resistance was observed with repeated exposure [28]. Dynamic interactions between the nanoparticles and the antimicrobial peptides afford enhanced stability and biological activity via a conjugation mechanism [32]. These findings provide essential insight into a potential strategy for overcoming antibiotic resistance by using ampicillin-AgNP conjugates as highly potent antimicrobial agents.

However, prior research has largely overlooked the cytotoxicity of these conjugates in mammalian cell lines, which is a key determinant for their safety and clinical applicability. Another critical gap in previous research is the lack of mutant prevention studies, which are essential for evaluating the potential of antimicrobial agents to suppress the emergence of resistant bacterial populations. Although the high antimicrobial potency of nanoparticle-antibiotic conjugates, involving antibiotics, is established [27,29], the accurate quantification of ampicillin conjugated per AgNP, which is critical for understanding the stability, have not been fully elucidated. Moreover, mechanistic insights into how these conjugates protect antibiotics from β-lactamase activity and inhibit biofilm, remain unexplored.

This study provides detailed optimization of the synthesis and characterization of AgNP-ampicillin conjugates to improve their activity against MRSA and other drug-resistant pathogens. By measuring the extent of conjugation and evaluating the ability of these conjugates to prevent mutation and biofilm formation and protect ampicillin from degradation by β-lactamases, this work seeks to provide a basis for clinically interchange systems that combine nanoparticles and antibiotics. The study also outlined the chemical synthesis process of ampicillin-AgNPs conjugates, offering a scalable method for developing nanoparticle-based antimicrobial agents. Besides, through the assessment of key metrics such as time-kill kinetic assay and FIC indices in addition to the cytotoxicity test, this research establishes the safety, efficacy, and potential of AgNP-ampicillin conjugates for use against resistant pathogens, addressing an important gap in nanotechnology-based antimicrobial therapy.

## 2. Materials and methods

### 2.1. Chemicals and reagents

Silver nitrate (AgNO3) (ACS Reagent, ≥ 99.0% pure) was obtained from Honeywell Fluka, NC, USA; (3-aminopropyl) triethoxysilane (APTES) (Analytical Grade, ≥ 98.0% pure) and ampicillin (Analytical Grade, ≥ 98.0% pure) were obtained from Alfa Aesar Haverhill, MA, USA. Sodium borohydride (ACS Reagent, ≥ 98.0% pure), tetraethyl orthosilicate (TEOS) (Analytical Grade, ≥ 99.0% pure), N-hydroxysuccinimide (NHS) (Analytical Grade, 98.0% pure) and 1-ethyl-3-(3-dimethylaminopropyl) carbodiimide (EDC) (Analytical Grade, ≥ 97.0% pure) were collected from Sigma–Aldrich, MO, USA.

Hydrogen peroxide (ACS Reagent, 30.0% pure) and trisodium citrate (ACS Grade, ≥ 99.0% pure) were purchased from Carl Roth, Karlsruhe, Germany. Ethanol (ACS Reagent, 99.5% pure) and ammonium hydroxide (ACS Reagent, 28.0–30.0% pure) were purchased from Merck, NJ, USA. Mueller-Hinton agar (MHA) (Analytical Grade, 99.0% pure) and Mueller-Hinton broth (MHB) (Analytical Grade, 99.0% pure) used for the antibacterial activity assays were both purchased from HiMedia Laboratories Pvt. Ltd., Mumbai, India. All reactions were carried out with deionized water, which was obtained from the Purelab water purification system, ELGA LabWater, UK.

### 2.2. Synthesis and purification methods of ampicillin-conjugated AgNPs

**2.2.1. Synthesis and purification of AgNPs.** An aqueous solution combining 10 mL silver nitrate (2.94 mM) and 30 mL trisodium citrate (2.91 mM) was stirred for 3 minutes along with the gradual addition of 50 µL hydrogen peroxide (30%). Twenty milliliters of sodium borohydride solution (5.87 mM) was injected into the solution mixture to initiate the reaction. The solution color changed from light yellow to reddish yellow after 3–4 minutes of continuous stirring [33]. The synthesized AgNPs were centrifuged at 14,000 rpm for 15 minutes at 4°C, and the supernatant was discarded. The AgNP pellets were washed twice with 80 mL of ethanol to remove unreacted reagents and impurities. After each washing step, the solution was centrifuged again at 14,000 rpm for 15 minutes at 4°C to collect purified AgNPs, which was stored between 2–8°C in a refrigerator.

In the synthesis of silver nanoparticles (AgNPs), silver nitrate (AgNO$_3$) serves as the precursor by providing silver ions (Ag$^+$), which are reduced to metallic silver (Ag$^0$) to form nanoparticles. Trisodium citrate, acting as both a reducing and capping agent, facilitates the reduction of Ag$^+$ into Ag$^0$ while simultaneously stabilizing the nanoparticles by preventing aggregation. Sodium borohydride (NaBH$_4$), a potent reducing agent, is used to accelerate the reduction process, ensuring rapid nanoparticle formation and uniform size distribution. Additionally, hydrogen peroxide (H$_2$O$_2$) serves as a stabilizing agent, moderating the reaction kinetics and enhancing the stability of the synthesized nanoparticles. Together, these reagents synergistically contribute to the efficient synthesis and stabilization of AgNPs, ensuring their suitability for subsequent functionalization and conjugation processes.

**2.2.2. Synthesis and purification of silica-coated silver nanoparticles (AgNP@SiO$_2$).** A total of 400 µL TEOS (98%), a silica precursor, was mixed with 5 mL ethanol under stirring for 5 minutes, and then the AgNP suspension

containing 5 mg AgNPs in 500 μL ammonia solution (25%) was added. The mixture was stirred for 12 hours at room temperature to adhere the silica shell on the surface of the AgNPs [34,35]. Afterwards, the synthesized AgNP@SiO$_2$ was centrifuged at 15,000 rpm for 10 minutes at 4°C, and the supernatant was discarded. The AgNP@SiO$_2$ pellets were washed thrice with 80 mL of ethanol to remove unreacted TEOS and ammonia completely. After each washing step, the solution was centrifuged at 15,000 rpm for 10 minutes at 4°C to collect purified AgNP@SiO$_2$.

Ammonia (NH$_3$) in the form of ammonium hydroxide solution acts as a catalyst in the Stöber process for silica shell formation around AgNPs by facilitating the hydrolysis of tetraethyl orthosilicate (TEOS) into silicic acid and subsequent condensation reactions to form a uniform silica shell on the surface of AgNPs. The basic environment provided by ammonia increases the reaction rate, ensuring efficient nucleation and growth of the silica layer. This silica coating stabilizes the nanoparticles and provides a functional surface for amine functionalization, enabling effective conjugation with ampicillin.

**2.2.3. Synthesis and purification of amine-functionalized silver nanoparticles (AgNP@SiO$_2$-NH$_2$).** Purified AgNP@SiO$_2$ was dispersed into a solvent mixture containing 20% water, 40% dimethyl sulfoxide (DMSO) and 40% ethanol along with 100 μL APTES (0.47 M) and subjected to sonication at a frequency of 40 kHz and power of 150 W for 5 minutes using a probe sonicator accompanied by an overnight reaction of 12 hours under continuous stirring [36]. The synthesized AgNP@SiO$_2$-NH$_2$ was centrifuged at 14,000 rpm for 15 minutes at 4°C, and the supernatant was discarded. The pellets were washed three times with 80 mL ethanol, with each washing step followed by centrifugation at 14,000 rpm for 15 minutes at 4°C, to collect purified AgNP@SiO$_2$-NH$_2$.

**2.2.4. Synthesis and purification of ampicillin-conjugated AgNPs (AgNP@SiO$_2$-NH-Amp).** Ethanolic solution (10 mL) of EDC (3.4 mM) was mixed with a 10 mL aqueous solution of ampicillin (3.4 mM) on a magnetic stirrer for 1 hour, which was followed by the addition of a 10 mL ethanolic solution of NHS (3.4 mM) under continuous stirring for another 1 hour at room temperature. Afterwards, AgNP@SiO$_2$-NH$_2$ was reacted overnight with this mixture under constant stirring at room temperature [29].

The synthesized AgNP@SiO$_2$-NH-Amp, which has lesser water solubility than free ampicillin, was centrifuged at 10,000 rpm for 10 minutes at 4°C, and the supernatant containing unbound ampicillin was removed. The pellets were then solubilized in 20 mL of ethanol and centrifuged at 10,000 rpm for 10 minutes at 4°C to remove any residual free ampicillin, which is practically insoluble in ethanol and forms a precipitate. The cycle was repeated three times, and the supernatant containing purified AgNP@SiO$_2$-NH-Amp was stored at 2–8°C. The supernatant from the purification steps was analyzed using UV-visible spectroscopy at 216 nm to confirm the absence of free ampicillin in the purified product.

For effective carbodiimide-mediated coupling, a molar ratio of 1:1 (EDC:NHS) was maintained. EDC activates the carboxyl group on ampicillin, forming an O-acylisourea intermediate, which is stabilized by NHS to generate an active NHS ester and subsequently reacts with the amine groups on AgNP@SiO$_2$-NH$_2$ to form stable amide bonds. The 1:1 ratio ensures sufficient activation of carboxyl groups while preventing excessive hydrolysis of intermediates, which could reduce conjugation efficiency.

**2.2.5. Quantification of unreacted ampicillin in the supernatant.** To confirm the efficiency of ampicillin conjugation with AgNPs, the concentration of unreacted ampicillin in the supernatant from the initial purification step was quantified using a UV-Visible spectrophotometer (T60, PG Instruments, UK). After the conjugation reaction, the supernatant was collected by centrifugation at 10,000 rpm for 10 minutes at 4°C and diluted with deionized water to a suitable concentration. The absorbance of the supernatant was recorded at 216 nm and compared with a standard calibration curve of pure ampicillin in deionized water. Calibration was performed in the 10–100 μg/mL range to obtain a calibration curve. The concentration of unreacted ampicillin was calculated from the absorbance values, and the conjugation efficiency (*E*%) was determined using the following equation:

$$E\,(\%) = \left(1 - \frac{\textit{Absorbance of unreacted ampicillin in the supernatant}}{\textit{Absorbance of initially added ampicillin}}\right) \times 100$$

***Equation 1: Equation for determining the conjugation efficiency***

The method used for quantifying conjugation efficiency is novel due to its ability to precisely calculate the percentage of ampicillin molecules conjugated to AgNPs using UV-Vis spectroscopy at 216 nm and further validated by TGA. Besides, EDX data can be utilized to determine the amount of silver present in the conjugate. This level of detail in quantification has not been consistently addressed in previous studies, providing a significant improvement in understanding the stability and functionalization of AgNP-antibiotic conjugates.

## 2.3. Chemical characterization methods of AgNPs and conjugated products

### 2.3.1. UV-Vis spectroscopy.
The optical absorption spectra for AgNPs in colloidal preparation were recorded using a double-beam UV-Visible spectrophotometer (T60, PG Instruments, UK) in the wavelength range of 200–600 nm, which ensures coverage of the characteristic surface plasmon resonance (SPR) peak of AgNPs (380–440 nm). Measurements were carried out with a spectral resolution of 1 nm to ensure precise peak detection using a quartz cuvette with a path length of 1 cm.

The SPR peak of AgNPs (380–440 nm) was measured to confirm the successful synthesis and stability of the nanoparticles. While SPR was not used to directly measure binding affinities, it played a critical role in validating the structural integrity of AgNPs for subsequent conjugation with ampicillin.

### 2.3.2. Dynamic Light Scattering (DLS) and Zeta Potential Measurements.
The size and size distribution of AgNPs and conjugated products were measured by DLS measurement instrument (Malvern, UK). The same equipment was also used to measure the zeta potential ($\zeta$) developed due to the surface charge of AgNPs. The electrophoretic mobilities of the particles were calculated using the Huckel equation.

AgNP solutions were analyzed in deionized water to minimize interference from other ions or stabilizing agents. On the other hand, Silica-coated AgNPs (AgNP@SiO$_2$), amine-functionalized AgNPs (AgNP@SiO$_2$-NH$_2$), and AgNP-ampicillin conjugates (AgNP@SiO$_2$-NH-Amp) were dispersed in ethylene glycol before measurement to ensure proper dispersion and minimize aggregation due to their modified surface chemistries.

Polydispersity index (PDI) were measured by dynamic light scattering (DLS) to quantify the uniformity of particle size distribution in nanoparticle samples. In nanoparticle research, a low PDI indicates a homogeneous (monodisperse) population, which is desirable for consistent performance and stability. PDI values typically range from 0.0 (perfectly uniform) to 1.0 (highly polydisperse), with values below 0.3 generally considered acceptable for nanoparticles conjugated with drug [37].

### 2.3.3. Scanning Electron Microscopy (SEM) and Energy Dispersive X-ray (EDX) Analysis.
The morphology of freeze-dried AgNPs, silica-coated AgNPs (AgNP@SiO$_2$), amine-functionalized AgNPs (AgNP@SiO$_2$-NH$_2$), ampicillin-conjugated AgNPs (AgNP@SiO$_2$-NH-Amp), and pure ampicillin was carried out using a JEOL analytical scanning electron microscope (SEM), model JSM-7600F.

In order to preparing the sample, nanoparticle suspensions were freeze-dried using a TOPT-10C Vacuum Freeze Dryer to remove solvents and obtain dry powders while preserving the nanoparticle morphology. Approximately 10 mL of each sample was frozen using liquid nitrogen and then subjected to freeze-drying at a chamber temperature of −50°C and a vacuum pressure of 0.1 mbar for 24 hours. The freeze-dried powders were stored in airtight containers to prevent moisture contamination.

A small amount of freeze-dried nanoparticle powder was evenly dispersed onto carbon adhesive tape affixed to aluminum SEM stubs. The excess powder was carefully removed using a soft brush to avoid contamination of the sample stub. The samples were sputter-coated with a thin layer of gold (~10 nm thickness) using a Quorum Q150R ES sputter coater to prevent charging effects and improve image quality during SEM analysis. The sputtering process was conducted under an argon atmosphere at 20 mA for 60 seconds.

SEM imaging was conducted using a JEOL JSM-7600F Analytical Scanning Electron Microscope under high vacuum conditions. The accelerating voltage was set between 5 and 15 kV, and magnifications ranged from 5,000× to 30,000× to analyze particle morphology, size, and surface structure.

EDX spectroscopy was conducted on the freeze-dried ampicillin-conjugated AgNPs (AgNP@SiO$_2$-NH-Amp) using the same SEM instrument equipped with an EDX detector to confirm the presence of silver (Ag) and other elements associated with functionalization and conjugation.

**2.3.4. Fourier Transform Infrared Spectroscopy (FTIR) analysis.** FTIR analysis of pure ampicillin and ampicillin-conjugated AgNPs (AgNP@SiO$_2$-NH-Amp) was carried out to confirm the functional groups and molecular interactions in ampicillin-conjugated AgNPs.

The samples were prepared using the pelletization method, where 1 mg of the dried sample was mixed with 100 mg of spectroscopic-grade potassium bromide (KBr) and ground into a fine powder using an agate mortar and pestle. The mixture was compressed into a thin pellet using a hydraulic press under a force of 10 tons for 5 minutes, which ensured uniform sample distribution and minimized scattering effects during analysis.

The FTIR spectra were recorded using an IR Prestige FTIR/NIR spectrometer (Shimadzu, Japan) in the wavelength range of 4000–400 cm$^{-1}$ with a resolution of 4 cm$^{-1}$ and a scanning speed of 2 mm/s. Baseline correction was applied to minimize noise and improve the accuracy of the spectra using the software's internal functions before peak assignment and integration. The recorded spectra were analyzed using Shimadzu IRsolution software (version 1.50) for peak identification, baseline correction, and data processing.

**2.3.5. Thermogravimetric Analysis (TGA).** Thermogravimetric analysis (TGA) of pure ampicillin and ampicillin-conjugated AgNPs (AgNP@SiO$_2$-NH-Amp) was performed to evaluate the thermal stability and extent of conjugation in ampicillin-conjugated AgNPs. The initial mass of each freeze-dried conjugated sample was precisely measured as 32 mg (containing ampicillin-conjugated AgNPs and other by-products) using an analytical balance (±0.1 mg accuracy) to ensure consistency across all measurements.

TGA was conducted using a Hitachi instrument (TG/DTA 7200, Japan) at a heating rate of 10°C per minute in the 20–800°C temperature range to monitor thermal degradation patterns. The analysis was run in a nitrogen atmosphere with a 50 mL/min flow rate to prevent oxidation during heating and ensure accurate thermal decomposition profiles. The weight-loss data were processed using TA7000 software (Hitachi, Japan) while baseline correction was applied to remove noise and ensure accurate interpretation.

## 2.4. Cytotoxicity assay

**2.4.1. Collection of vero cell.** The kidney epithelial cell from African green monkeys (Vero cell line) were sourced from the Cell and Tissue Culture Laboratory, Centre for Advanced Research in Sciences (CARS), University of Dhaka, Bangladesh.

**2.4.2. Culture of vero cell.** Vero cells were cultured in Dulbecco's Modified Eagle's Medium (DMEM) supplemented with 10% fetal bovine serum (FBS), 1% penicillin-streptomycin (1:1), and 0.2% gentamycin. Cells were maintained at 37°C in a humidified atmosphere containing 5% CO$_2$. The medium was replaced every 3 days, and cells were subcultured upon reaching 80–90% confluence using trypsin-EDTA for detachment [38].

After incubation at 37°C for 3 minutes, detached cells were collected, centrifuged at 1000 rpm for 5 minutes, resuspended in fresh medium, and seeded into new culture flasks at the appropriate split ratio [38]. All procedures were performed under sterile conditions to ensure cell viability and minimize contamination.

**2.4.3. Cytotoxicity evaluation of test samples.** The cytotoxicity of the test samples to Vero cells was studied in triplicate (S1 = Ampicillin, S2 = AgNP-ampicillin conjugate, S3 = AgNO$_3$, and S4 = AgNPs) at the Centre for Advanced Research in Sciences (CARS), University of Dhaka, Bangladesh.

The cells were plated in a 96-well plate at a density of 1.5 × 10$^4$ cells/100 μL per well and were incubated at 37°C with 5% CO$_2$ for 24 hours. Following this, a volume of 25 μL from each autoclaved test sample with a concentration of 100 μg/mL was added in duplicate into the respective wells and incubated for 24 hours. Cell survival percentages were measured by microscopic examination using an inverted light microscope after incubation [39].

### 2.5. Microbiological characterization of AgNO₃, AgNPs, AgNP-ampicillin with pure ampicillin

The antibacterial activity of $AgNO_3$, AgNPs, AgNP-ampicillin, and pure ampicillin was evaluated in four ampicillin-resistant (*Bacillus subtilis*, *Escherichia coli*, *Staphylococcus aureus, and Pseudomonas aeruginosa*) clinical isolates using different microbiological assays (Table 1).

**2.5.1. Collection of clinical isolates.** The study included ampicillin-resistant clinical isolates of four bacterial species: *Bacillus subtilis*, *Escherichia coli*, *Staphylococcus aureus*, and *Pseudomonas aeruginosa*, which were sourced from microbiology laboratory of Mohakhali TB Hospital and BIRDEM Medical College (Table 2). These bacteria were chosen because of their clinical significance, their resistance profiles, and their capacity for biofilm formation, which renders them difficult to treat with conventional antibiotics. However, detailed clinical data, including the primary disease and treatment response, were not available due to ethical restrictions.

The clinical isolates used in this study were collected during routine diagnostic microbiology procedures and were anonymized before use. Ethical approval was obtained from the Ethical Review Committee, Faculty of Pharmacy, University of Dhaka, and all experiments were conducted following the Declaration of Helsinki and relevant guidelines.

**2.5.2. General preparation of media and controls for microbiological assays.**

*Preparation of microbial culture media*: To prepare MHA media, 19.5 g of MHA powder was dissolved in distilled water to a 500 mL volume with heat and agitation, followed by the adjustment of pH to $7.3 \pm 0.1$ using 0.1M NaOH. The solution was autoclaved at 15 psi and 121°C for 15 minutes and cooled to 45–50°C. For solid media, 15 mL of autoclaved MHA was poured into sterile petri dishes and allowed to solidify at room temperature for 1 hour, followed by storage at 4–8°C.

To make MHB media, 10.5 g of Mueller-Hinton Broth (MHB) was mixed with distilled water, heated, and adjusted to a 500 mL volume. After that, the pH of the medium was adjusted to $7.3 \pm 0.1$ using 0.1M NaOH. The medium was autoclaved at 15 psi and 121°C for 15 minutes and later reduced to 45–50°C before storing in a refrigerator at 4–8°C to maintain sterility. All media and agar plates were thawed to room temperature before use.

**Table 1. Testing method and objective of microbiological test.**

| Test Name | Testing Method | Objective | References |
|---|---|---|---|
| **Disc diffusion test** | Spread plating method | Qualitative assay by zone of inhibition | [40] |
| **MIC test** | Broth dilution method | Quantitative assay against planktonic cells | [41] |
| **MBC test** | Spread plating method | Quantitative assay against planktonic cells | [41] |
| **MPC test** | Agar dilution- drop plating method | Evaluation antibiotic resistance development | [42,43] |
| **MBIC test** | Crystal violet biomass staining method | Quantitative assay against adherent cells | [44,45] |
| **MBEC test** | Crystal violet biomass staining method | Quantitative assay against adherent cells | [44,45] |
| **Time-kill kinetics assay** | Broth dilution- drop plating method | Evaluation of Antimicrobial efficacy | [46,47] |
| **FIC index** | Checkerboard testing method | Antimicrobial synergy study | [48,49] |

**Table 2. List of sample-resistant bacterial strains for microbiological testing.**

| Bacteria Name | Gram stain | Respiration | Collection Source |
|---|---|---|---|
| *Bacillus subtilis* (*B. subtilis*) | Gram-positive | Aerobic | Mohakhali TB Hospital |
| *Escherichia coli* (*E. coli*) | Gram-negative | Aerobic | BIRDEM Medical College |
| *Staphylococcus aureus* (*S. aureus*) | Gram-positive | Facultative anaerobic | BIRDEM Medical College |
| *Pseudomonas aeruginosa* (*P. aeruginosa*) | Gram-negative | Facultative anaerobic | Mohakhali TB Hospital |

***Preparation of bacterial inoculation:*** Bacterial inoculum was prepared by transferring a single bacterial colony from a fresh culture plate into 5 mL of sterile saline (0.85% NaCl), followed by standardization to a 0.5 McFarland turbidity standard (equivalent to approximately $1.5 \times 10^8$ CFU/mL) using a spectrophotometer to match an optical density (OD) of 0.08–0.13 at 600 nm to ensure consistent inoculum density across all experiments.

***Preparation of master stock solution:*** The starting concentration (500 µg/mL) of test samples (AgNO₃, AgNPs, AgNP-ampicillin, and pure ampicillin) was set by dissolving 5 mg (5000 µg) of test samples in 10 mL sterile deionized water, since 5 mg is the least measurable amount on a four-decimal-place balance with reasonable accuracy. The master stock solution was then prepared by serially diluting each initial sample solution with sterile MHB media (1:2 dilution series) to achieve a concentration of 125 µg/mL in the final volume of 20 mL. Master stocks were prepared separately in the same experimental condition for conducting MIC, MPC, MBIC, MBEC, and Time-kill kinetic assays.

***Preparation of working test solutions:*** The cytotoxicity studies demonstrated that little or no toxicity occurred in Vero cells up to 100 µg/mL of test samples (AgNO₃, AgNPs, AgNP-ampicillin, and pure ampicillin) and established the safety profile towards human cells. Therefore, 100 µg/mL was chosen as the starting working concentration to evaluate the antibacterial activity, which was prepared by mixing 8 mL of the stock solution with 2 mL of MHB. Afterwards, 50 µg/mL, 25 µg/mL, 12.5 µg/mL, and 6.25 µg/mL concentrations of the test samples were prepared by doubling dilutions of the 100 µg/mL concentrated solution. These adjusted dilutions ensured that all working test solutions would maintain the consistent concentrations in the MIC, MBC, MPC, MBIC, MBEC, and Time-kill kinetic assays.

***Preparation of controls:*** For the growth or positive control, 2 mL of sterile MHB was inoculated with 20 µL of a standardized bacterial suspension and incubated at 37°C for 20 hours to confirm the viability of bacterial strains and the suitability of the media.

Sterility or negative controls consisted of 2 mL of sterile MHB without bacterial inoculum or test compounds, incubated under identical assay conditions, verifying the absence of contamination in the media or setup.

Solvent controls were prepared with 1 mL sterile deionized water (used to make initial test sample solutions) added to 1 mL MHB to ensure that the solvent alone did not influence bacterial growth. These controls established baselines for comparison and validated the experimental results.

**2.5.3. Disk diffusion test.** The disk diffusion test was carried out to measure the zone of inhibition guided by CLSI M02 Performance Standards for Antimicrobial Disk Susceptibility Tests (2018) (Table 2) [40,50]. To confirm the reproducibility of the results, three biological replicates were performed for each test sample and solvent control (deionized water) using bacterial cultures derived from independent subcultures for each bacterial strain.

The standardized bacterial suspension was spread evenly on the solidified MHA plates using a sterile cotton swab, and the plates were dried for 1 hour before placing the prepared disks. Filter disks (6 mm diameter) were impregnated with 20 µL of test sample solutions (containing 10 µg per disk) along with solvent, air-dried for 30 minutes to ensure uniform sample distribution, and placed on the solidified agar surface using sterile forceps.

After that, the plates were incubated invertedly at 37°C for 20 hours in 5% $CO_2$ to ensure complete bacterial growth and clear zone formation. However, a shorter incubation time (e.g., 16–18 hours) could be sufficient for rapidly growing organisms. After incubation, the diameter of the inhibition zones was measured in millimeters using a Vernier caliper.

***Statistical analysis:*** The statistical analysis was carried out using IBM SPSS Statistics (version 29), and results were presented as mean ± standard deviation. One-way ANOVA was performed for mean diameters of inhibition zones (mm) of three treatments (AgNPs, ampicillin, AgNP-ampicillin) for each clinical isolate with three biological replicates. Since silver nitrate demonstrated no inhibition in the testing concentrations, they were not considered for further statistical analysis to avoid statistical error. One-way ANOVA tests the null hypothesis, assuming that all sample groups' means are equal. Tukey's HSD post-hoc test was used to determine the pairwise differences.

***One-way ANOVA procedure:*** The data were initially checked for normal distribution and equal variance. When these criteria were satisfied, one-way ANOVA was performed to compare the means between the four treatment groups. From

the test, an F-statistic and a p-value were obtained. A p-value <0.05 was considered significant, meaning that at least two group means differed significantly.

*Tukey's Post Hoc test procedure*: After obtaining a significant ANOVA result, Tukey's Honestly Significant Difference (HSD) test was applied as a post hoc analysis to pinpoint specific pairwise differences between treatment groups. Tukey's HSD controls the family-wise error rate and is suitable when the assumption of homogeneity of variances is met. The test presented adjusted p-values for pairwise comparisons, and $p < 0.05$ was considered significant.

**2.5.4. MIC, MBC and MPC assay.** The minimum inhibitory concentration (MIC) and minimum bactericidal concentration (MBC) were determined to measure the activity of samples against planktonic bacterial cells using CLSI M07 Methods for Dilution Antimicrobial Susceptibility Tests for Bacteria (2018) (Table 2) [41,51].

To determine the MIC of each test sample against different bacterial isolates, 20 µL of inoculum was added to each test tube, except the sterility control. The MIC was measured after incubation at 37°C for 20 hours in 5% $CO_2$, by comparing test tubes to sterility controls to identify the lowest concentration inhibiting bacterial growth.

To evaluate the MBC of test samples, 10 µL from the MIC dilution suspension and the preceding dilutions were transferred into separate screw cap tubes for each bacterium, then adjusted to 10 mL with sterile MH broth. From these, 100 µL was spread on agar plates using a swab and incubated at 37°C for 16 hours in 5% $CO_2$ to observe for bacterial colony presence. The lowest concentration with no visible growth represents the minimum bactericidal concentration (MBC), indicating a 99.9% reduction of the initial bacterial inoculum.

The lowest concentration of test samples that prevents the growth of least susceptible, single-step resistant mutants within bacterial isolates was evaluated by mutant prevention concentration (MPC) assay as described by Mani *et al.* (2006) and Blondeau *et al.* (2001), with slight modifications in plating techniques (Table 2) [42,43].

To assess the MPC of test samples, 3 mL of solutions, at concentrations equivalent to the minimum bactericidal concentration (MBC) and the preceding dilutions, were added to separate test tubes with 12 mL of sterile media, followed by solidification at 25°C for 1 hour and 4–8°C for 2 hours.

200 µL of inoculum was added to each test tube except for the sterility control, containing sterile 10 mL MHB media, followed by 16 hours of incubation at 37°C in 5% $CO_2$. After incubation, cultures were estimated to have a $\geq 3 \times 10^8$ CFU/mL measured by a UV-visible spectrophotometer, which provided an optical density (OD) of 0.256–0.261 at 600 nm. Cultures were concentrated by centrifugation at 5000 rpm for 30 minutes and resuspended the pellet thoroughly in 0.3 mL of sterile MHB media using a vortex to obtain a concentration of around $10^{10}$ CFU/mL [52].

Afterward, 50 µL of bacterial cell suspension was dropwise (10 µL/drop) applied to the agar plate containing test samples. Inoculated plates were incubated for 16 hours at 37°C in 5% $CO_2$ and then screened for growth. MPC was recorded as the lowest test sample concentration, which allowed no growth. The minimum preventive concentration (MPC) was determined by identifying the concentration that did not allow bacterial growth.

**2.5.5. MBIC and MBEC assay.** Anti-biofilm activities of the samples against adherent bacterial cells were examined by minimum biofilm inhibitory concentration (MBIC) and minimum biofilm eradication concentration (MBEC) following the modified method of Thieme *et al.* (2016), Costa *et al.* (2018) (Table 2) [44,45]. While the MBIC assay evaluated the inhibition of bacterial biofilm formation, the MBEC method focused on removing the preformed biofilm by the test samples. Standardized Bacterial inoculum, Media and controls were prepared as described in the section 2.5.2. All absorbance readings were averaged over three technical replicates to ensure statistical reliability and consistency of the results.

For the MBIC assay, 2 mL of each test sample dilution was inoculated with 20 µL of the standardized bacterial suspension and kept at 37°C for 24 h for biofilm formation. Following incubation, the test tubes were gently rinsed with sterilized deionized water to remove any planktonic cells while keeping the biofilm intact, which was adhered to the walls of the test tube. Adherent biofilm cells were stained with 3 mL of 0.1% crystal violet solution before washing with sterile deionized water and then dissolving in 3 mL of 96% ethanol. The absorbance of the crystal violet-ethanol solution was measured at 590 nm using a spectrophotometer with sterility control as a blank. MBIC was determined as the minimum concentration

of the test sample that produced an absorbance less than 0.1, indicating significant inhibition of biofilm formation. On the contrary, absorbance values above 0.1 indicated insufficient inhibition of biofilm growth.

For the MBEC assay, biofilms were first allowed to form during the 24-hour incubation of 2 mL sterile MHB inoculated with 20 µL of a standardized bacterial suspension at 37°C. After washing with sterile deionized water to remove planktonic cells while keeping the biofilm attached to the sides of the test tube, 2 mL of dilutions of each test sample was added to the biofilm-containing test tubes. The tubes were incubated for another 4 hours at 37°C to evaluate the eradication of preformed biofilms under the favorable growing conditions for bacteria. The tubes were then washed with sterilized deionized water, stained with 0.1% crystal violet, and processed according to the protocol for the MBIC assay. The MBEC was determined to be the lowest concentration of the test sample that can kill all microorganisms within a biofilm, which is associated with biofilm disaggregation effect.

**2.5.6. Time-kill kinetic assay.** Time-kill kinetics assay reveal the interactions of $AgNO_3$, AgNPs, AgNP-ampicillin, and pure ampicillin with the clinical isolates at different time intervals. The assay investigated the concentration or time-dependent impact of $AgNO_3$, AgNPs, AgNP-ampicillin and pure ampicillin on bacterial strains. The time-kill kinetic assay was performed according to the guidelines of the ASTM Standard Guide for Assessment of Antimicrobial Activity Using a Time-Kill Procedure (2016) (Table 2) [46,47]. Standardized bacterial inoculum, media and controls were prepared as described in the section 2.5.2.

In this assay, 2 mL of each test sample dilution were placed in screw cap glass vials and inoculated with 20 µL of a standardized bacterial suspension to achieve a cell density around $1.5 \times 10^6$ CFU/mL in the final volume. The 20 µL volume was optimized to ensure consistent bacterial growth while maintaining a proportional bacterial load for accurate time-kill measurements. The vials containing test samples, growth, solvent and sterility controls were incubated at 37°C for 8 hours in 5% $CO_2$ under shaking conditions, with samples collected at 0, 2, 4, 6, and 8 hours. At each time point, serial dilutions of the samples as well as controls were performed to obtain a range of concentrations suitable for counting [53]. Each sterile agar plate was segmented into four quadrants designated for a specific dilution in the series.

For this purpose, after vortexing the vials for around 5s, 10 µL of the suspension was withdrawn and replaced by sterile MHB to ensure constant volume. The withdrawn suspension was then serially diluted with sterile MHB up to four times. After that, 50 µL of the diluted suspension was spread onto sterile MHA plates using the drop-plate method, with 50 µL divided into five evenly spaced 10-µL drops. The process was repeated for each dilution of the dilution series. The countable dilution should provide 3–30 colonies per 10-µL drop of the dispensed material, which aligns with the SP method requiring counting at a sample dilution comprising 30–300 CFUs per plate [54]. These MHA plates were prepared in triplicate to ensure reproducibility of the colony counts. Finally, plates were incubated at 37°C for 16 hours in 5% $CO_2$ for proper bacterial colony development and accurate CFU enumeration.

The Leica Darkfield Quebec Colony Counter was employed to count the colonies over time with proper magnification of the drops. The total count of CFUs was recorded and scaled up to express the viable cell counts in CFU/mL. After that, the log10 reduction in CFU/mL was calculated over time. The bactericidal activity was assessed by determining the $\log_{10}$(CFU/mL) reduction at 0th and 8th hours, with ≥3 $\log_{10}$(CFU/mL) decrease indicating effective bactericidal action against all bacterial strains [55].

*Calculation methods for time-kill kinetics*

$$N\,(CFU/ml) = \frac{\overline{n} \times F \times 1000\ (\mu L)}{V\ (\mu L)}$$

Where, N = Number of colonies at time in CFU/mL at time t in hour, $\overline{n}$ = Average number of colonies per drop, F = Dilution factor, V = Drop volume in µL.

$$log\,R = log\,N_0 - log\,N_8$$

$$\text{Or,}\;\; log\,R = \frac{log\,N_0}{log\,N_8}$$

***Equation 2: Equation for determining log10 reduction in the number of bacterial colonies***
Where, $log\,R$ = log10 reduction of colonies in CFU/mL, $log\,N_0$ = Initial number of colonies in CFU/mL, and $log\,N_8$ = Final number of colonies in CFU/mL at 8h.
   ***Interpretations of time-kill kinetics***

$log\,R \geq 3$: This suggests strong bactericidal activity

$1 \leq log\,R \leq 3$: This denotes moderate bactericidal activity

$log\,R < 1$: This points towards weak bactericidal activity

   **2.5.7. Fractional inhibitory concentration (FIC) index.** The FIC index was determined after measuring the MIC values of the pure ampicillin, AgNPs, and AgNP-ampicillin combination by checkerboard assay, then calculating and interpreting the results according to the literature [48,56] (Table 2). The MIC values of ampicillin and AgNPs were determined separately before evaluating the FIC index following standard protocols (as described in the MIC section). The FIC index was determined to evaluate the interaction between ampicillin and AgNPs in the AgNP-ampicillin conjugates, quantifying the synergistic effect of the combination. The FIC index value quantifies the interaction by summing the ratios of each compound's concentration in the combination to its individual MIC values, as calculated using the formula:

$$\text{FIC Index} = \text{MIC}_A + \text{MIC}_B$$

   ***Equation 3: Equation for determining FIC index***
   Where:
   $\text{FIC}_A$ = A/ $\text{MIC}_A$ (Concentration of ampicillin in the combination divided by the MIC of ampicillin alone)
   $\text{FIC}_B$ = B/ $\text{MIC}_B$ (Concentration of AgNPs in the combination divided by the MIC of AgNPs alone)
***Interpretation of FIC values:*** In the current study, the synergisms are interpreted based on the criteria devised by Odds *et al.* (2003) [57] (Table 3).
   **FIC ≤ 0.5**: Synergistic effect. This suggests that the combination of ampicillin and AgNPs is more effective than either agent alone, thereby enhancing the inhibition of bacterial growth
   **0.5 < FIC ≤ 4**: No interaction. This indicates that the combination does not significantly improve upon the individual effects, rendering their interaction neutral.
   **FIC > 4**: Antagonistic effect. This refers that the combination of ampicillin and AgNPs is less effective than either agent alone, suggesting that one compound may diminish the efficacy of the other.

**Table 3. Interpretation of the FIC index.**

| Interpretation | FIC Index |
|---|---|
| Synergistic effect | ≤0.5 |
| No interaction | >0.5-4 |
| Antagonistic effect | >4.0 |

## 2.6. Molecular docking

Molecular docking was performed for ampicillin and AgNP-ampicillin with Beta-lactamase enzyme (PDB ID: 1XPB) using AutoDock Vina (1.2.0) [58,59] and visualized the results using Biovia Discovery Studio (v21.1.0.20298) [60]. The process initiated with the preparation of the Beta-lactamase protein by downloading its structure from the RCSB Protein Data Bank in PDB format, water molecules removed and hydrogen atoms were added. The refined protein was then saved in PDBQT format.

Ampicillin and AgNP-ampicillin were modeled and optimized for the ligands using molecular editor Avogadro (version: 1.2.0) [61], then converted to PDBQT format as well. The AgNP-ampicillin ligand contains two silicon (Si) atoms and one silver (Ag) atom. However, AutoDock is not well-suited to handle uncommon metals like Ag (Silver) or Si (Silicon) without some adjustments. Therefore, silicon (Si) was replaced with carbon (C) and silver (Ag) with hydrogen (H) to run docking using AutoDock Vina.

The grid box for docking was configured to encompass the active site of Beta-lactamase, specifying center coordinates and dimensions. Docking was performed using AutoDock Vina, with parameters set for exhaustiveness and the number of output conformations. Post-docking, Biovia Discovery Studio was employed to visualize and analyze the binding poses, interactions, and energies.

The results revealed key insights into the binding affinities and interactions of both compounds with β-lactamase, highlighting the potential advantages of AgNP-ampicillin in overcoming β-lactamase-mediated resistance. Ligand binding affinity (kcal/mol) indicated the binding energy between ligand and protein; a more negative binding affinity value denoted a stronger binding. The Root Mean Square Deviation upper bound (RMSD/ub) was computed to evaluate the deviation of the predicted ligand pose from the reference structure, and a higher RMSD value represents less similarity. While showing alignment accuracy, we used the Root Mean Square Deviation lower bound (RMSD/lb), where a lower value indicated greater fit of the ligand in the receptor binding site. These parameters were important in assessing the validity of the docking results and probable functional activity of the ligands towards the inhibition of β-lactamase.

## 3. Results

### 3.1. Chemical characterization of AgNPs, ampicillin, and AgNP-ampicillin

#### 3.1.1. Characterization of synthesized silver nanoparticles.
Due to surface plasmon resonance (SPR), AgNPs absorb light radiation in the range of 380 nm to 440 nm (Fig 1A) and produce a characteristic color that confirms nanoparticle synthesis [62]. This phenomenon is dependent on the interaction between the free electrons in the AgNPs and the electromagnetic field of the visible light. SPR also provides insights into the stability of the nanoparticles. The presence of a sharp and well-defined absorption peak suggests that the AgNPs are well-dispersed, with no significant aggregation. FTIR spectra of AgNPs observed at 3409 cm$^{-1}$ are indicative of the stretching vibration of the hydroxyl group of trisodium citrate (Fig 1B) that reduced the Ag$^+$ ions as well as capped the reduced AgNPs [63]. As the melting point of AgNPs (i.e., 112 °C) is less than the melting point of silver (i.e., 961.8 $^0$C), the TGA curve (Fig 1C) shows characteristic mass loss patterns consistent with the formation of AgNPs [64].

The size distribution curve (Fig 1D) of AgNPs measured by dynamic light scattering (DLS) provides the particle size, i.e., Z-Average is 67.34 nm in diameter with polydispersity index (PDI) of 0.316 indicating moderate polydispersity. The zeta potential distribution curve (Fig 1E) of AgNPs measured by electrophoretic light scattering (ELS), i.e., The Z-Average of −36.7 mV indicates a better stability profile, as values more negative than −30 mV suggest enhanced stability [65]. The SEM image (Fig 1F) provides visual confirmation of the size and morphology of the synthesized AgNPs, showing spherical particles with uniform distribution, while the DLS analysis (Fig 1D) provides a statistical distribution of particle sizes in suspension, with a Z-average diameter of 67.34 nm. Together, these techniques confirm the successful synthesis of AgNPs with consistent size and shape.

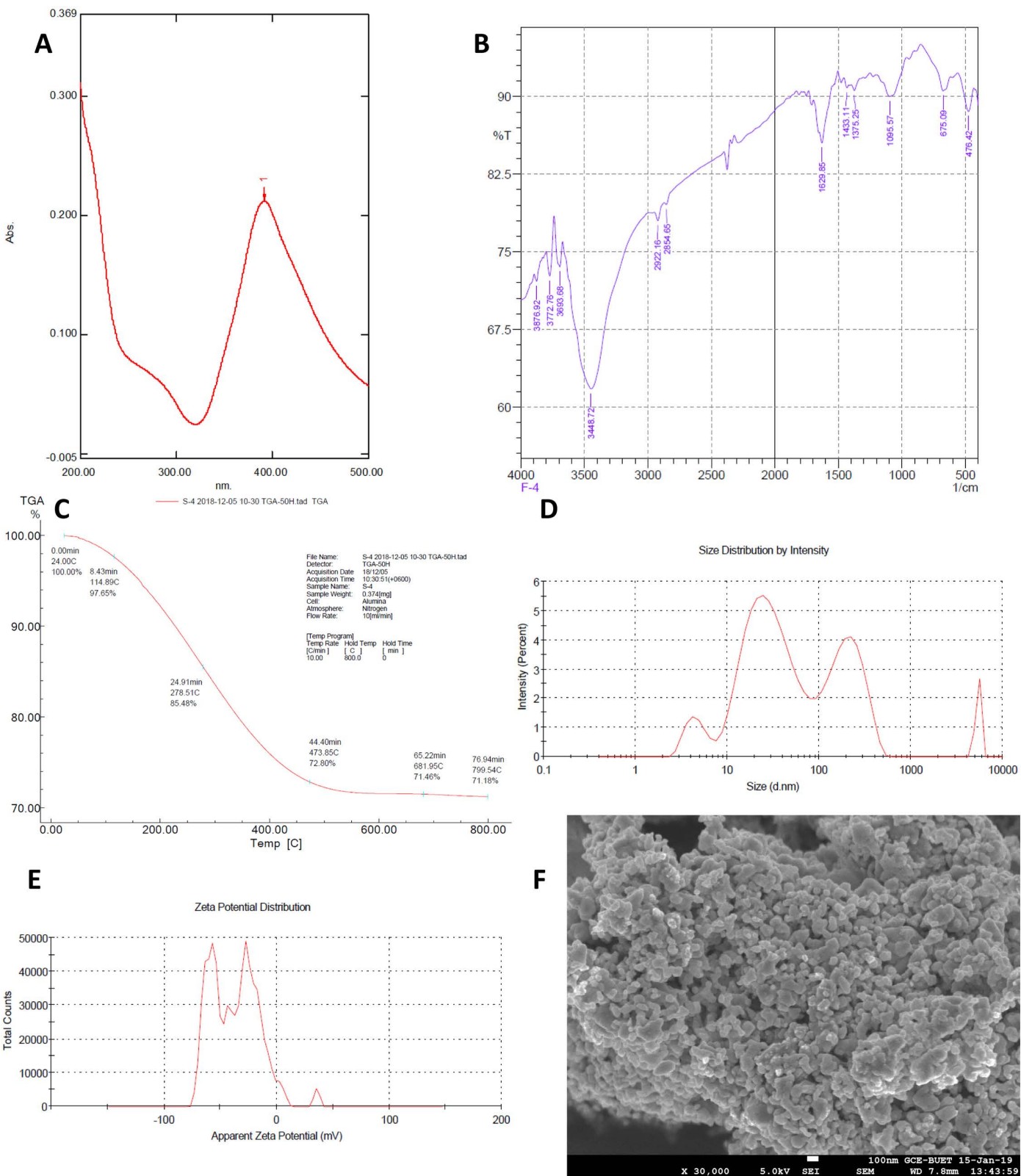

**Fig 1. Characterization of synthesized silver nanoparticles; (A) UV–visible absorption spectra.** (B) FTIR spectra. (C) TGA in the temperature range of 20–800°C. (D) Size distribution by intensity. (E) Zeta potential distribution. (F) SEM image (×30000).

### 3.1.2. Characterization of silica-coated and amine-functionalized silver nanoparticles.

Reduced degradation in TGA (Fig 2A) was observed due to the silica coating of AgNPs [66], and the size distribution curve (Fig 2B) of silica-coated AgNPs measured by dynamic light scattering (DLS) confirms the increase in particle size, i.e., Z-Average is 253.2 nm in diameter with PDI value of 0.300. The SEM image (Fig 2C) also confirms the size increase in silica-coated AgNPs. Increase weight loss in the TGA curve (Fig 3A) was observed due to amine functionalization of AgNPs [66] and the size distribution curve (Fig 3B) of amine functionalized AgNPs measured by dynamic light scattering (DLS) confirms the increase in particle size, i.e., Z-Average is 578.6 nm in diameter with PDI value of 0.092 ensuring monodisperse particle synthesis. The SEM image (Fig 3C) also confirms the size increase in amine-functionalized AgNPs.

### 3.1.3. Characterization of AgNP-ampicillin conjugate against pure ampicillin.

In the FTIR data of pure ampicillin (Fig 4A), the carboxylic group of ampicillin exhibits a robust broad band between 2600 cm$^{-1}$ and 3200 cm$^{-1}$ for the hydroxyl stretch. The bands at 1693 cm$^{-1}$ and 1583 cm$^{-1}$ correspond to carbonyl stretching vibrations of the lactam ring and primary amine, respectively. The peaks at 1518 cm$^{-1}$, 1383 cm$^{-1}$ and 1308 cm$^{-1}$ contributed to characteristic β lactam ring vibrations [28]. In the FTIR data of AgNP-ampicillin (Fig 4B), AgNP-ampicillin shows a robust broad band in the range of 3300 cm$^{-1}$ to 3500 cm$^{-1}$ for the N-H stretch, and the presence of only one spike confirms secondary amide formation. The peak at 1639 cm$^{-1}$ validates the formation of secondary amines in AgNP-ampicillin since the peak of primary amines is present at 1583 cm$^{-1}$ in pure ampicillin. The β-lactam ring bands (1772 cm$^{-1}$) of AgNP-ampicillin remain indistinguishable from those of pure ampicillin. The peak at 1113 cm$^{-1}$ in AgNP-ampicillin confirms the stretching vibration of the silyl ether group, which supports the conjugation between tetraethyl orthosilicate (TEOS) and (3-aminopropyl) triethoxysilane (APTES) in amine-functionalized AgNPs [67]. TEOS serves as the silica precursor, forming a silica shell around the nanoparticles, while APTES introduces amine groups to the silica surface for further functionalization and conjugation with ampicillin.

The TGA of AgNP-ampicillin (Fig 4D) reveals three different weight losses. In the temperature region of 25–160 °C, 4.54% weight loss contributes to the physical desorption of ampicillin from the surface of AgNP-ampicillin. In the temperature region of 161–644 °C, the weight loss is indicative of higher energy requirements for the desorption of ampicillin from AgNPs owing to the covalent interaction of an amine group in ampicillin with AgNPs. At a higher temperature region of approximately 644–800 °C, 2.2% weight loss may correspond to the van der Waals interaction mediated by the amine groups [28]. Comparing the curves of AgNP-ampicillin and pure ampicillin, observed between 160.78°C and 339.21°C (Fig 4C), confirms the 57.686% presence of ampicillin in AgNP-ampicillin, as calculated using conjugation efficiency (E%) equation (Equation 1).

The SEM images provided detailed insights into the morphology, confirming particle shapes, sizes, and the impact of silica coating, amine functionalization, and ampicillin conjugation. The interparticle distance among ampicillin molecules was observed in the SEM image of ampicillin (Fig 4E), whereas the SEM image of AgNP-ampicillin (Fig 4F) provides a more compact and larger structure without interparticle distance, which confirms the conjugation. The size distribution by intensity (Fig 5A) confirms the increase in size of conjugates, i.e., Z-Average of 1185 nm in diameter with PDI value of 0.096 confirms homogeneity, whereas the zeta potential of −35.7 mV (Fig 5B) indicates the optimal stability of AgNP-ampicillin.

The EDX spectra of the synthesized AgNP-ampicillin conjugates confirmed the presence of silver (Ag) along with peaks for carbon (C), oxygen (O), nitrogen (N), and silicon (Si). These elements validated the successful conjugation of ampicillin and the silica coating on the nanoparticles. Quantitative EDX analysis revealed the average silver content in the AgNP-ampicillin conjugates to be 1.52%, consistent across multiple regions of the sample. (Fig 5C–F) [68].

### 3.1.4. Yield calculation.

The average yield of the conjugated product was 32 milligrams (Table 4), and the theoretical amount of AgNP-ampicillin present in the conjugated product was equivalent to 12 milligrams of pure ampicillin since 12 mg of ampicillin were weighted to prepare 10 mL of a 3.4 mM ampicillin solution for conjugation reaction (Table 4).

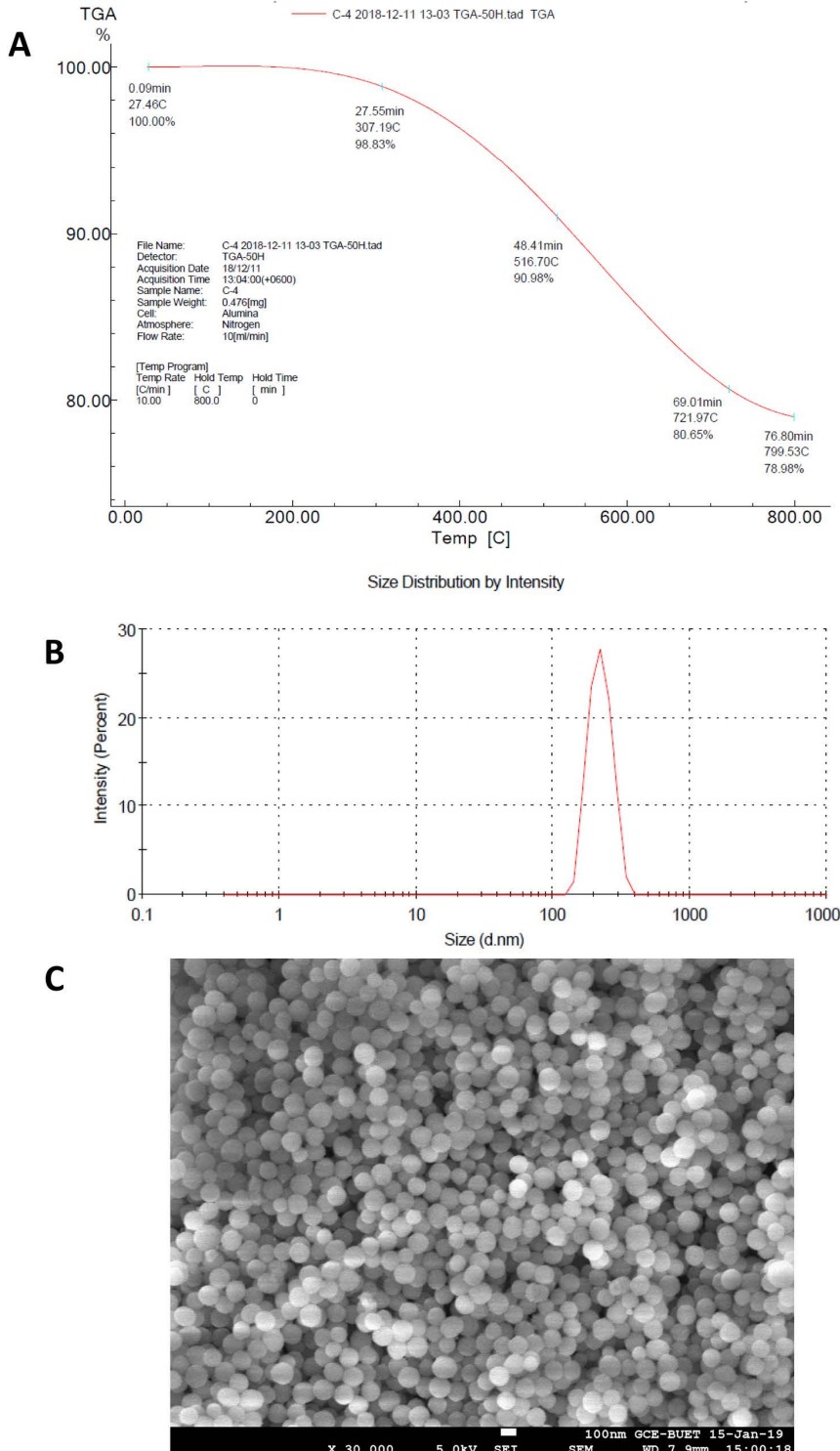

**Fig 2. Characterization of silica-coated silver nanoparticles; (A) TGA in the temperature range of 20–800°C.** (B) Size distribution by intensity. (C) SEM image (×30000).

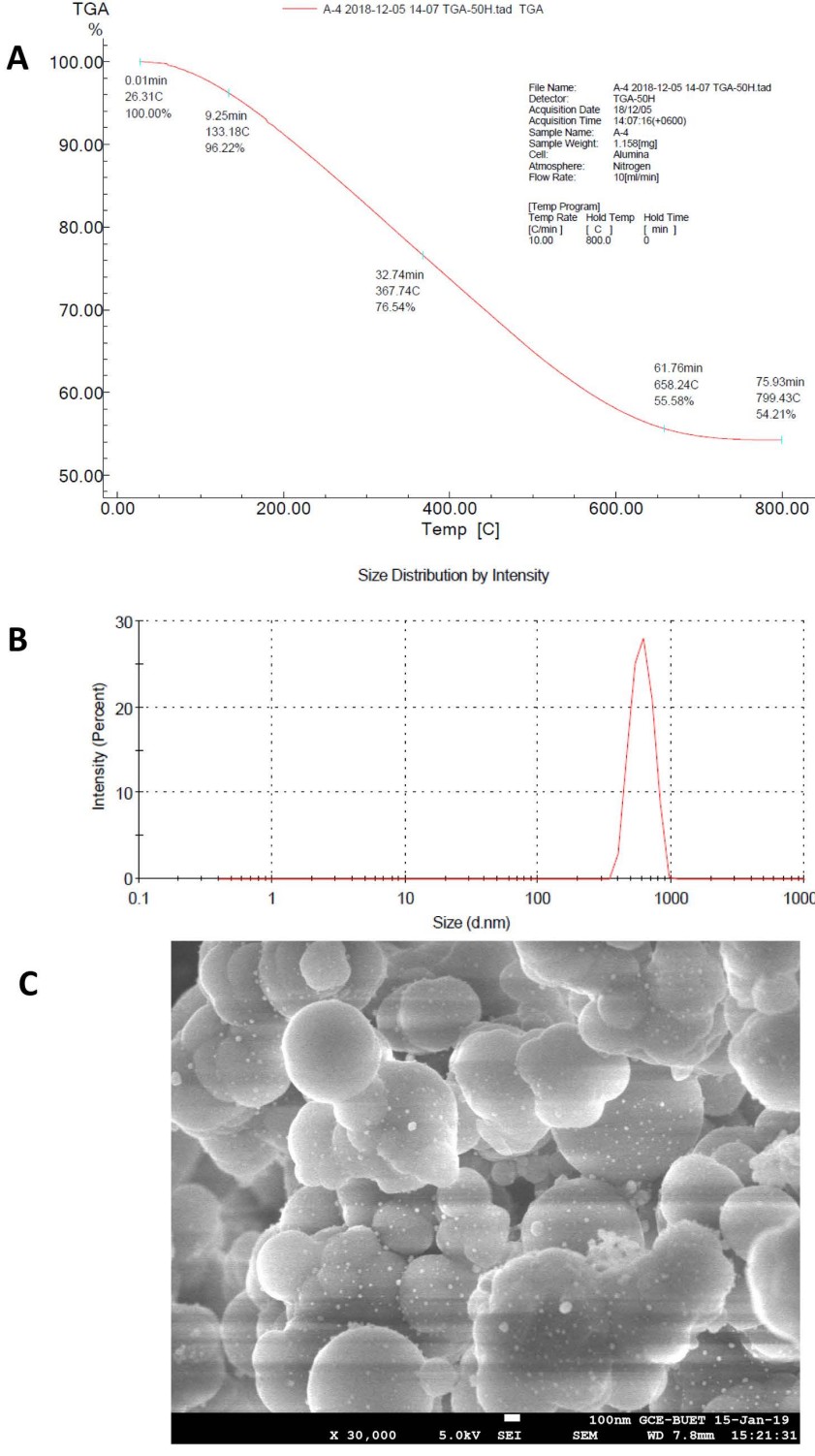

**Fig 3. Characterization of amine-functionalized silver nanoparticles; (A) TGA in the temperature range of 20–800°C.** (B) Size distribution by intensity. (C) SEM image (×30000).

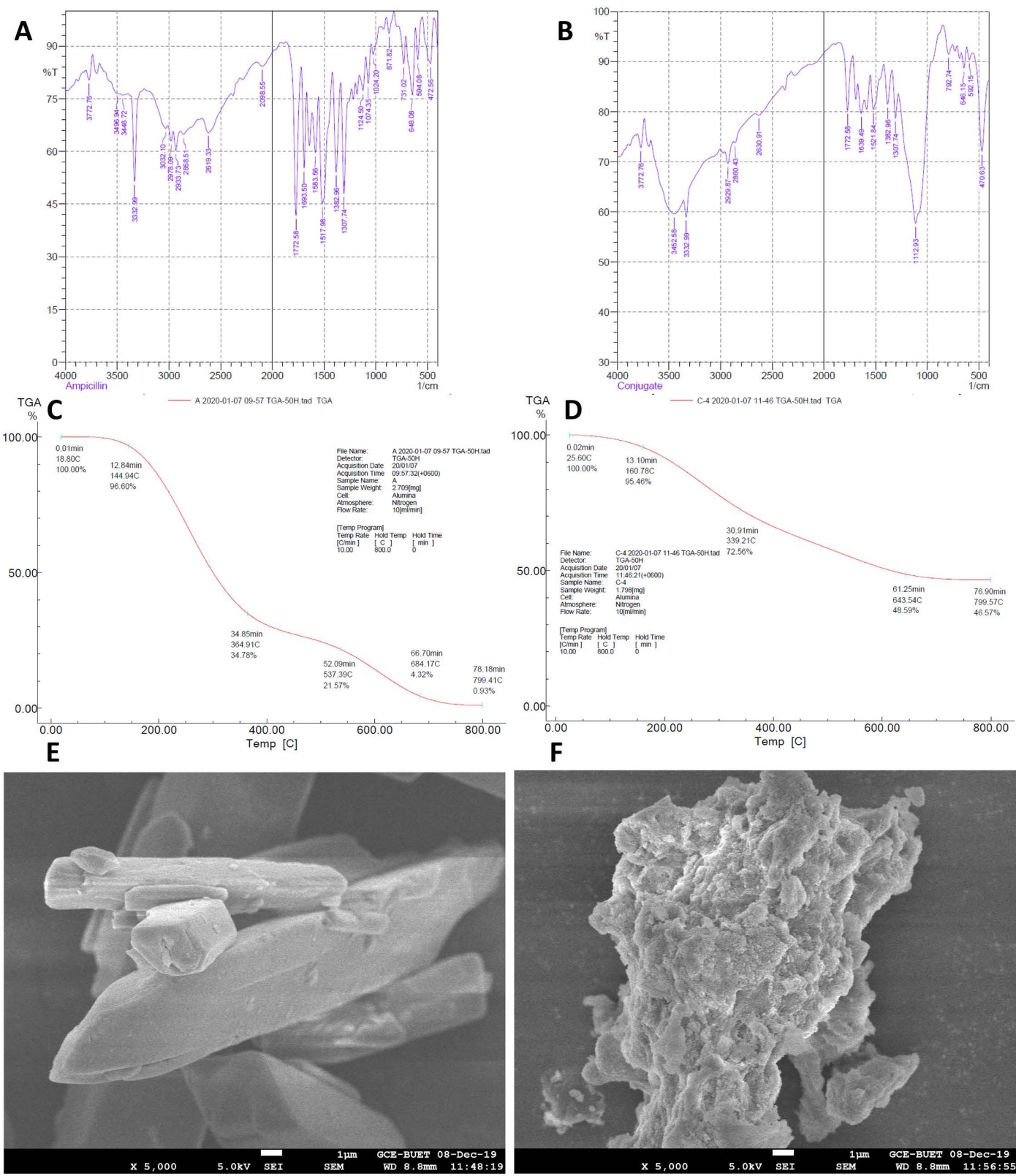

**Fig 4. Characterization of AgNP-ampicillin conjugate against pure ampicillin; (A) FTIR spectra of pure ampicillin.** (B) FTIR spectra of AgNP-ampicillin. (C) TGA of pure ampicillin in the temperature range of 20–800°C. (D) TGA of AgNP-ampicillin in the temperature range of 20–800°C. (E) SEM image of pure ampicillin (×5000). (F) SEM image of AgNP-ampicillin (×5000).

Calibration curve of pure ampicillin at 216 nm was obtained with an R² value of ≥0.9981, indicating a strong linear relationship between concentration and absorbance (S1 Figure in S1 File).

From the spectrophotometric data, absorbance of initially added ampicillin was 0.319, and absorbance of unreacted ampicillin in supernatant was 0.135. So, according to the equation-1, conjugation efficiency is 57.686%, which is equivalent to 57.7% (Equation 1).

Therefore, the effective quantity of ampicillin on AgNPs was found to be approximately 57.7%, which was also supported by the TGA curve (Fig 4C, D). Therefore, the practical amount of AgNP-ampicillin present in the conjugated product was equivalent to approximately 6.92 milligrams of pure ampicillin. In addition, it was estimated from the yield calculation that approximately four carboxyl group-functionalized ampicillin molecules are bound to one amine group-functionalized AgNP in an AgNP-ampicillin conjugate (Table 4).

### 3.2. AgNP-ampicillin synthesis reaction

In this study, synthesized silver nanoparticles (AgNPs) were coated with tetraethyl orthosilicate (TEOS) and subsequently amine functionalized using 3-amino propyl trimethoxysilane (APTES). On the other hand, carboxylic group of ampicillin were functionalized by using 1-Ethyl-3-(3-dimethyl aminopropyl) carbodiimide (EDC) and N-Hydroxysuccinimide (NHS) respectively. Afterwards, through the reaction of functionalized AgNP and functionalized ampicillin, AgNP-ampicillin conjugate containing an amide linkage were yielded. According to the yield calculation, four ampicillin molecules bound to each functionalized AgNP in the conjugate.

The conjugation reaction involves two key steps: (1) silica coating and amine functionalization of AgNPs, where TEOS forms a silica shell around the AgNPs, and APTES introduces amine groups on the shell surface; and (2) carbodiimide-mediated conjugation of ampicillin to the amine-functionalized nanoparticles, forming stable amide bonds. This detailed mechanism is illustrated in Figs 6 and 7.

### 3.3. Cytotoxicity assay

The cytotoxicity assay results provided important insight regarding the safety of AgNP-ampicillin conjugate. Although and silver nitrate (N1) and silver nanoparticles (N2) display potential cytotoxic effects with reduced cell survival (80–90%), the ampicillin-AgNPs conjugation seems to reduce these effects, as can be seen by the high cellular survival (>95%) observed for the AgNPs-ampicillin conjugate (S2), same as the survival of pure ampicillin (S1) (Table 5). This lower toxicity is due to the "shielding effect" of the antibiotic molecules that may prevent direct interaction of this nanoparticle with mammalian cell membranes. The presence of silica coating and amine functionalization during synthesis will help further improve the biocompatibility of the conjugates (Fig 8).

Substantial cell survival percentages for the AgNP-ampicillin conjugate (S2) also demonstrated the evidence for the AgNP-ampicillin conjugate as a candidate therapeutic against ampicillin-resistant infections. These findings were also consistent with the broader aim of the study, which was to develop a broad range of nanoparticle-based formulations of antibiotics with improved antibacterial activity and reduced cytotoxicity.

### 3.4. Microbiological characterization of AgNO₃, AgNPs, ampicillin, and AgNP-ampicillin

**3.4.1. Disk diffusion assay.** The disk diffusion test was performed against the ampicillin-resistant clinical isolates (Table 1), and the zone diameter in millimeters was determined (Table 6).

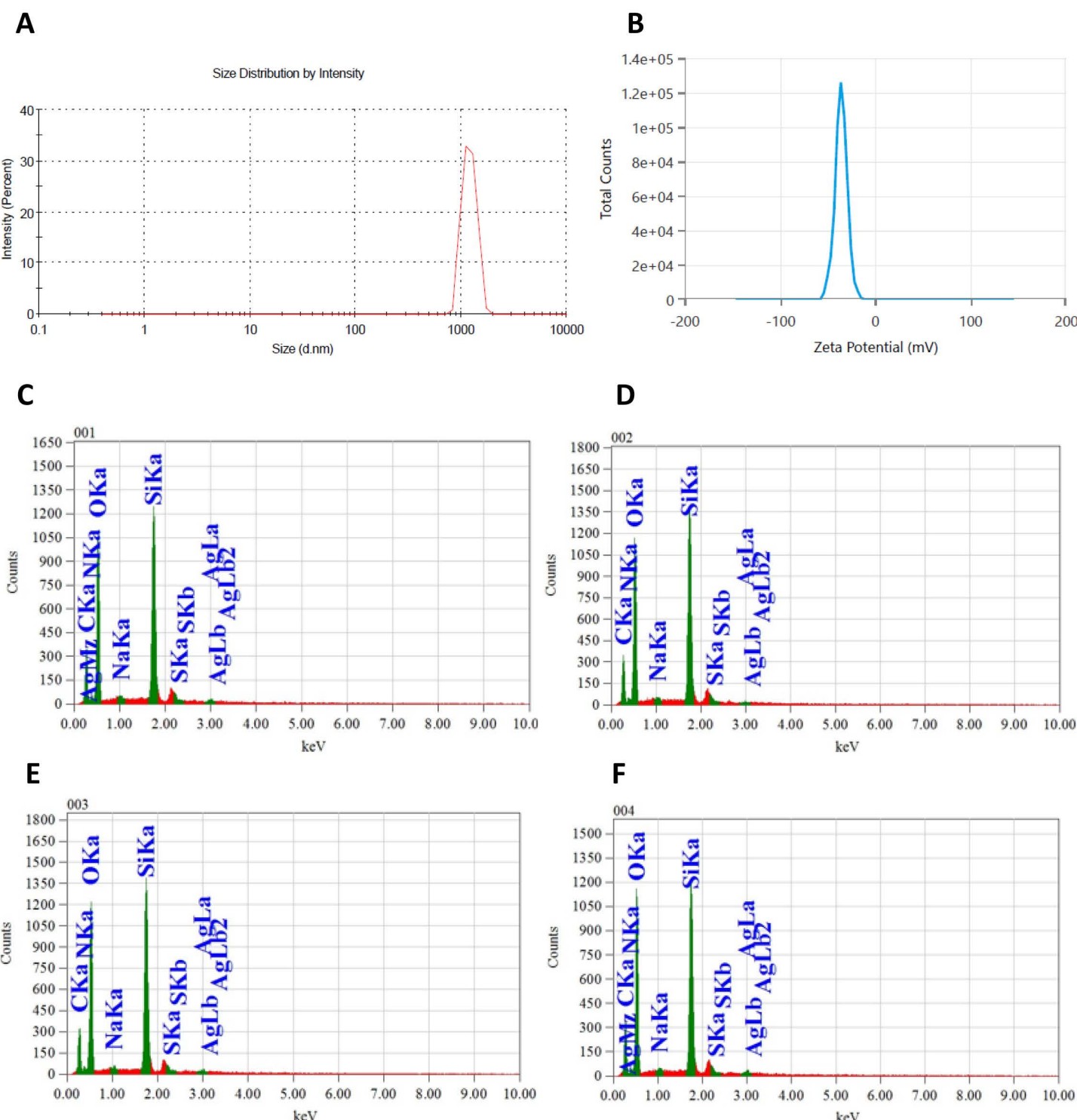

**Fig 5. Size distribution and EDX spectrum of the AgNP-ampicillin conjugate; (A) Size distribution by intensity.** (B) Zeta potential. (C) EDX Spectrum (Ag = 1.92%). (D) EDX Spectrum (Ag = 1.08%). (E) EDX spectrum (Ag = 1.49%). (F) EDX spectrum (Ag = 1.59%). Average Ag = 1.52%.

**Table 4. Calculation of the number of ampicillin molecules bound per AgNP.**

| Topic | Amount | Number of molecules |
|---|---|---|
| Average yield of conjugated product per batch | 32 mg | ----- |
| Theoretical amount of AgNP present in the conjugated product | 5 mg | $2.79179 \times 10^{19}$ molecules |
| Practical amount of AgNP present in the conjugated product | 32 mg × 1.52% (Fig 5C–F) = 0.4864 mg | $2.7159 \times 10^{18}$ molecules |
| Theoretical yield of Ampicillin conjugated with AgNP | 12 mg | $2.06854 \times 10^{19}$ molecules |
| Practical yield of Ampicillin conjugated with AgNP | 12 mg × 57.686% (Equation 1, Fig 4C, D) = 6.92232 mg | $1.19326 \times 10^{19}$ molecules |
| Number of Ampicillin molecule bound per AgNP | | ($1.19326 \times 10^{19}$ molecules)/ ($2.7159 \times 10^{18}$ molecules) = 4.4 ≈ 4 |

One way ANOVA indicated that the mean zones of inhibition differed significantly among treatments for all the clinical isolates tested (*B. subtilis*: F(2,6) = 25.68, p < 0.05; *E. coli*: F(2,6) = 18.00, p < 0.05; *S. aureus*: F(2,6) = 23.36, p < 0.05; *P. aeruginosa*: F(2,6) = 26.47, p < 0.05).

Tukey's HSD post-hoc testing illustrated that the combination of AgNP-ampicillin had a synergistic effect on most isolates. For *B. subtilis*, the combined treatment resulted in a much larger inhibition zone (15.83 ± 0.26 mm) than did AgNPs (11.83 ± 0.63; Δ 4.0 mm, p < 0.05) and ampicillin alone (13.50 ± 0.26 mm; Δ 2.33 mm, p < 0.05). Besides, for *E. coli*, the combination (13.83 ± 0.29 mm), was more effective than AgNPs (11.33 mm ± 1.05 mm; Δ = 2.5 mm, p < 0.05) and ampicillin alone (11.83 ± 0.77 mm; Δ = 2.0 mm, *p* < 0.05).

Similarly, for *Staphylococcus aureus*, AgNP-ampicillin combination (13.17 ± 0.29 mm) exhibited a significantly larger mean inhibition zone compared to both AgNPs (9.83 ± 0.76 mm; Δ = 3.34 mm, *p* < 0.05) and ampicillin alone (10.33 ± 0.76 mm; Δ = 2.84 mm, *p* < 0.05). Likewise, for *Pseudomonas aeruginosa*, AgNP-ampicillin combination (14.00 ± 0.50 mm) outperformed both AgNPs (10.83 ± 0.29 mm; Δ = 3.17 mm, *p* < 0.05) and ampicillin alone (11.67 ± 0.58 mm; Δ = 2.33 mm, *p* < 0.05).

Standard deviations indicated moderate technical variability, a known challenge in disk diffusion assays. Despite the small sample size (*n* = 3 biological replicates), consistent statistical significance across isolates underscores the robustness of the findings. The lack of inhibition by silver nitrate alone confirms that silver nanoparticles, rather than silver ions, drive the observed effects, aligning with prior studies on AgNP mechanisms.

**3.4.2. MIC, MBC and MPC assay.** The MIC, MBC and MPC values suggested increased antibacterial activity of AgNP-ampicillin in comparison to pure ampicillin, likely due to the synergistic interaction between silver nanoparticles (AgNPs) and ampicillin, which could improve bacterial cell wall penetration, reduce β-lactamase activity, and increase overall stability.

For all clinical isolates (Table 1), the lowest concentration of samples that inhibited bacterial growth (MIC value) was determined (Table 7), where the conjugated product exhibited lower MIC values (i.e., 6.25, 6.25, 6.25, 12.5 ppm) in comparison with AgNPs (i.e., 12.5, 12.5, 12.5, 25 ppm) and pure ampicillin (i.e., 12.5, 12.5, 12.5, 25 ppm) against the ampicillin-resistant clinical isolates of *Bacillus subtilis, Escherichia coli, Staphylococcus aureus,* and *Pseudomonas aeruginosa* respectively.

The concentration of sample where no visible growth refers to the concentration (MBC value) that almost totally (99.9%) killed the initial inoculum of the clinical isolates (Table 1) was determined (Table 7), where the conjugated product displayed lessened MBC values (i.e., 12.5, 12.5, 12.5, 25 ppm) contrasted to AgNPs (i.e., 25, 25, 50, 50 ppm) and pure ampicillin (i.e., 12.5, 25, 25, 50 ppm) against the ampicillin-resistant clinical isolates of *Bacillus subtilis, Escherichia coli, Staphylococcus aureus* and *Pseudomonas aeruginosa* correspondingly.

**Fig 6. Silica coating and amine functionalization reaction of AgNPs.**

The concentration of the sample where no mutation growth occurred after inoculation with approximately $10^{10}$ bacterial cells on agar plates was chosen as the MPC value (Table 7), where the conjugated product demonstrated ability to prevent the growth of resistant mutant at 50, 100, 100, 100 ppm concentration against the ampicillin-resistant clinical isolates of *Bacillus subtilis*, *Escherichia coli*, *Staphylococcus aureus* and *Pseudomonas aeruginosa* correspondingly. However, the pure ampicillin was found to be resistant against mutant *Staphylococcus aureus* and *Pseudomonas aeruginosa*.

**Fig 7. Carboxyl functionalization and conjugation of ampicillin with AgNPs.**

**Table 5. Cytotoxicity assay of test samples on Vero cell line.**

| Sample ID | Survival of Vero cells | Remarks |
|---|---|---|
| Negative Control (Only media) | 100% | No cytotoxicity |
| Solvent Control (Distilled water) | >95% | No cytotoxicity |
| N1 (AgNO$_3$) | 80-90% | Indicating cytotoxic effect |
| N2 (AgNPs) | 80-90% | Indicating cytotoxic effect |
| S1 (Ampicillin) | >95% | No cytotoxicity |
| S2 (AgNP-ampicillin) | >95% | No cytotoxicity |

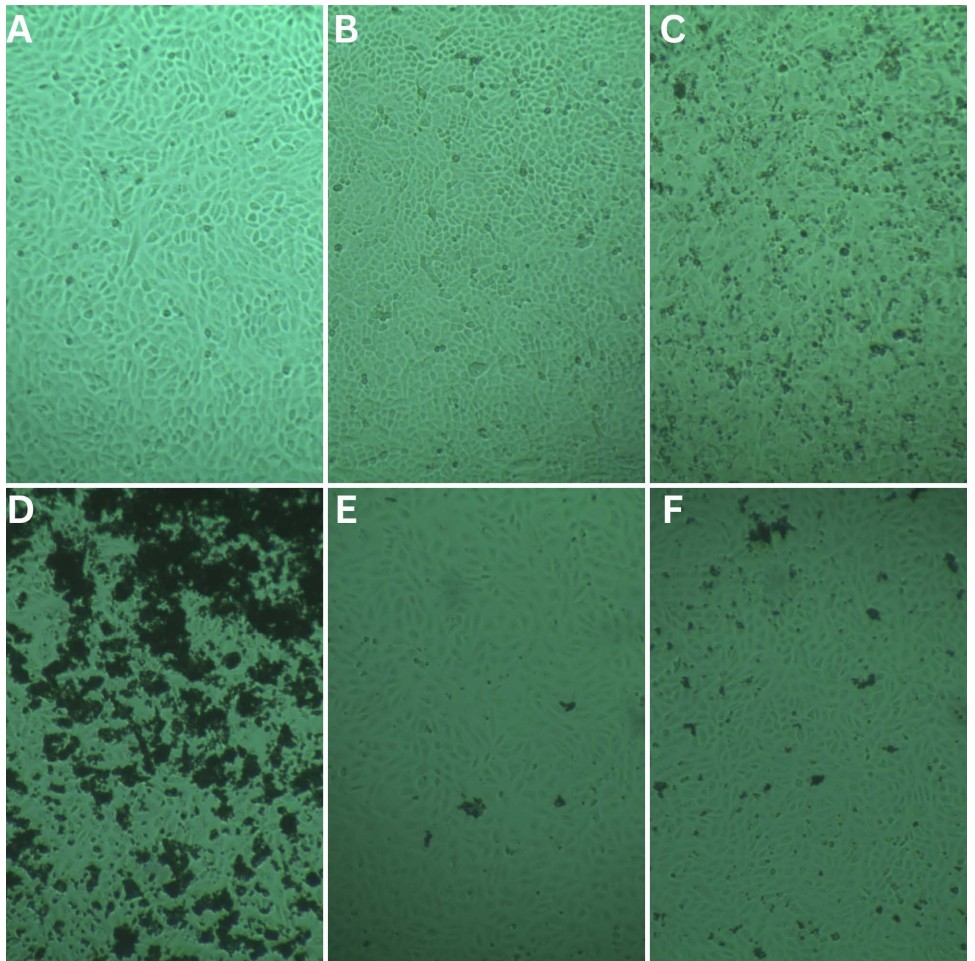

**Fig 8. Cytotoxicity assay of the test samples; (A) Negative Control.** (B) Solvent Control. (C) AgNO$_3$. (D) AgNPs. (E) Ampicillin. (F) AgNP-ampicillin.

**3.4.3. MBIC and MBEC assay.** MBIC was the lowest concentration of the sample (Table 8), which inhibited biofilm growth on the surface with an absorbance at 590 nm less than 0.1, where a sterility control was used as a blank. MBEC was the lowest concentration of the sample (Table 8), which removed the grown biofilm from the surface with an absorbance at 590 nm less than 0.1, where the sterility control was used as a blank. The MBIC and MBEC values stipulated a significant increase in the anti-biofilm activity of AgNP-ampicillin in contrast to pure ampicillin, which could

**Table 6. Antibacterial Activity of AgNPs, Ampicillin, and AgNP-Ampicillin Combination Against Clinical Isolates: Mean Zone of Inhibition (mm±SD) and ANOVA Results.**

| Clinical isolates | AgNPs (mm) | Ampicillin (mm) | AgNP-ampicillin (mm) | F-statistic (df) | p-value |
|---|---|---|---|---|---|
| *B. subtilis* | 11.83±0.63 | 13.50±0.26 | 15.83±0.26 | F (2,6) = 25.68 | p<0.05 |
| *E. coli* | 11.33±1.05 | 11.83±0.77 | 13.83±0.29 | F (2,6) = 18.00 | p<0.05 |
| *S. aureus* | 9.83±0.76 | 10.33±0.76 | 13.17±0.29 | F (2,6) = 23.36 | p<0.05 |
| *P. aeruginosa* | 10.83±0.29 | 11.67±0.58 | 14.00±0.50 | F (2,6) = 26.47 | p<0.05 |

**Table 7. MIC, MBC and MPC value of test samples in µg/mL displayed against resistant clinical isolates.**

| Clinical isolates | Assay | AgNO$_3$ | AgNPs | Ampicillin | AgNP-ampicillin |
|---|---|---|---|---|---|
| *B. subtilis* | MIC | >100 | 12.5 | 12.5 | 6.25 |
| | MBC | >100 | 25 | 12.5 | 12.5 |
| | MPC | >100 | 50 | 50 | 50 |
| *E. coli* | MIC | >100 | 12.5 | 12.5 | 6.25 |
| | MBC | >100 | 25 | 25 | 12.5 |
| | MPC | >100 | >100 | 100 | 100 |
| *S. aureus* | MIC | >100 | 12.5 | 12.5 | 6.25 |
| | MBC | >100 | 50 | 25 | 12.5 |
| | MPC | >100 | >100 | >100 | 100 |
| *P. aeruginosa* | MIC | >100 | 25 | 25 | 12.5 |
| | MBC | >100 | 50 | 50 | 25 |
| | MPC | >100 | >100 | >100 | 100 |

be attributed to the ability of AgNPs to interact with and penetrate the biofilm matrix, disrupting its structural integrity and enhancing the efficacy of ampicillin.

The conjugated product indicates better antibiofilm activity against the ampicillin-resistant clinical isolates of *Bacillus subtilis, Escherichia coli, Staphylococcus aureus,* and *Pseudomonas aeruginosa,* which is manifested by the MBIC values (i.e., 25, 50, 50, 100 ppm, respectively) and the MBEC values (i.e., 100, 100, 100, 100 ppm, respectively) of the conjugated product. In contrast, the pure ampicillin provided higher MBIC values (i.e., 50, 100, 100, 100 ppm) against the ampicillin-resistant clinical isolates of *Bacillus subtilis, Escherichia coli, Staphylococcus aureus* and *Pseudomonas aeruginosa* respectively. However, it could not eradicate the biofilm formed by *Staphylococcus aureus,* and *Pseudomonas aeruginosa.*

**3.4.4. Time-kill kinetic assay.** The time-kill kinetic assay demonstrated that AgNP-ampicillin exhibited faster and more effective antibacterial activity compared to pure ampicillin, achieving a 3.81-, 3.42-, 3.38-, and 3.56-log10 reduction in CFU/mL for *Bacillus subtilis, Escherichia coli, Staphylococcus aureus,* and *Pseudomonas aeruginosa* within 8 hours, compared to 3.43-, 3.05-, 2.75-, and 2.85-log10 reductions achieved by pure ampicillin. This indicates that AgNP-ampicillin works 1.11-, 1.12-, 1.23-, and 1.25-times faster than pure ampicillin in reducing bacterial populations of the same bacterial strains, respectively (Table 9). The time-kill curve of the test samples also reaffirmed the enhanced time-dependent activity of AgNP-ampicillin compared to the pure ampicillin (Fig 9).

**3.4.5. FIC index.** The FIC index value was used to quantify synergy and determine the impact of a combination of antibiotics on potency in contrast to their independent actions. The FIC value signals a synergistic effect of ampicillin and AgNP in the AgNP-ampicillin conjugate (Table 10).

**Table 8. MBIC and MBEC value of test samples in µg/mL showed against resistant clinical isolates.**

| Clinical isolates | Assay | AgNO₃ | AgNPs | Ampicillin | AgNP-ampicillin |
|---|---|---|---|---|---|
| *B. subtilis* | MBIC | >100 | 100 | 50 | 25 |
| | MBEC | >100 | >100 | 100 | 100 |
| *E. coli* | MBIC | >100 | >100 | 100 | 50 |
| | MBEC | >100 | >100 | 100 | 100 |
| *S. aureus* | MBIC | >100 | >100 | 100 | 50 |
| | MBEC | >100 | >100 | >100 | 100 |
| *P. aeruginosa* | MBIC | >100 | >100 | 100 | 100 |
| | MBEC | >100 | >100 | >100 | 100 |

**Table 9. logarithmic reduction (logR) in the initial number of colonies calculated using Equation 2 after 8 hours of incubation with test samples; (-): No Reduction.**

| Clinical isolates | AgNO₃ | AgNPs | Ampicillin | AgNP-ampicillin |
|---|---|---|---|---|
| *B. subtilis* | – | 2.33 | 3.43 | 3.81 |
| *E. coli* | – | 2.05 | 3.05 | 3.42 |
| *S. aureus* | – | 1.68 | 2.75 | 3.38 |
| *P. aeruginosa* | – | 1.72 | 2.85 | 3.56 |

The FIC value of 0.25 indicates a strong synergistic effect of ampicillin and AgNP against the ampicillin-resistant *Bacillus subtilis* and *Escherichia coli,* while the FIC value of 0.5 denotes a moderate synergistic effect of ampicillin and AgNP against the ampicillin-resistant *Staphylococcus aureus* and *Pseudomonas aeruginosa.*

### 3.5. Molecular docking

The docking results of Ampicillin and AgNP-ampicillin at Beta-lactamase (PDB ID: 1XPB) were quite diverse regarding binding affinity, pose stability, and structural interaction. The Ampicillin showed higher binding efficacy and extensive interaction with Beta-lactamase (Fig 10A), and AgNP-ampicillin possessed better pose stability and distinctive structural nature (Fig 10B). The findings suggest a potential mechanism by which AgNP-ampicillin conjugates evade β-lactamase degradation. However, these results are hypothetical and require further experimental validation.

**3.5.1. Binding affinity.** Ampicillin had binding affinity between −8.1 and −6.1 kcal/mol, while the highest being −8.1 kcal/mol. AgNP-ampicillin, on the other hand, manifested binding affinities between −7.5 and −6.3 kcal/mol and a best pose with an affinity of −7.5 kcal/ mol. This suggests that ampicillin has a higher binding affinity than AgNP-ampicillin and makes more favorable interactions with the active site of Beta-lactamase. Ampicillin may bind more tightly because it can establish more rigid and/or specific interactions with essential residues in the active site, as it has been well known that molecular recognition contributes toward ligand binding.

**3.5.2. Pose stability.** The stability of the pose was investigated based on the RMSD values. For the best affinity poses of both ligands, a low deviation (RMSD/ub and RMSD/lb = 0) was obtained, suggesting reliability in the docking predictions. However, ampicillin displayed more variability among the other poses, ranging in RMSD/ub out to 28.661 Å versus maximum RMSD/ub for AgNP-ampicillin of 8.381 Å, which may imply that although ampicillin offers greater binding affinity at its best pose, it is not the most stable. Overall, its binding throughout other conformations may be less uniform. AgNP-ampicillin displayed more stable binding across its poses together, likely due to the monomeric structure of the AgNP-ampicillin, allowing it to possess potentially interesting interaction properties.

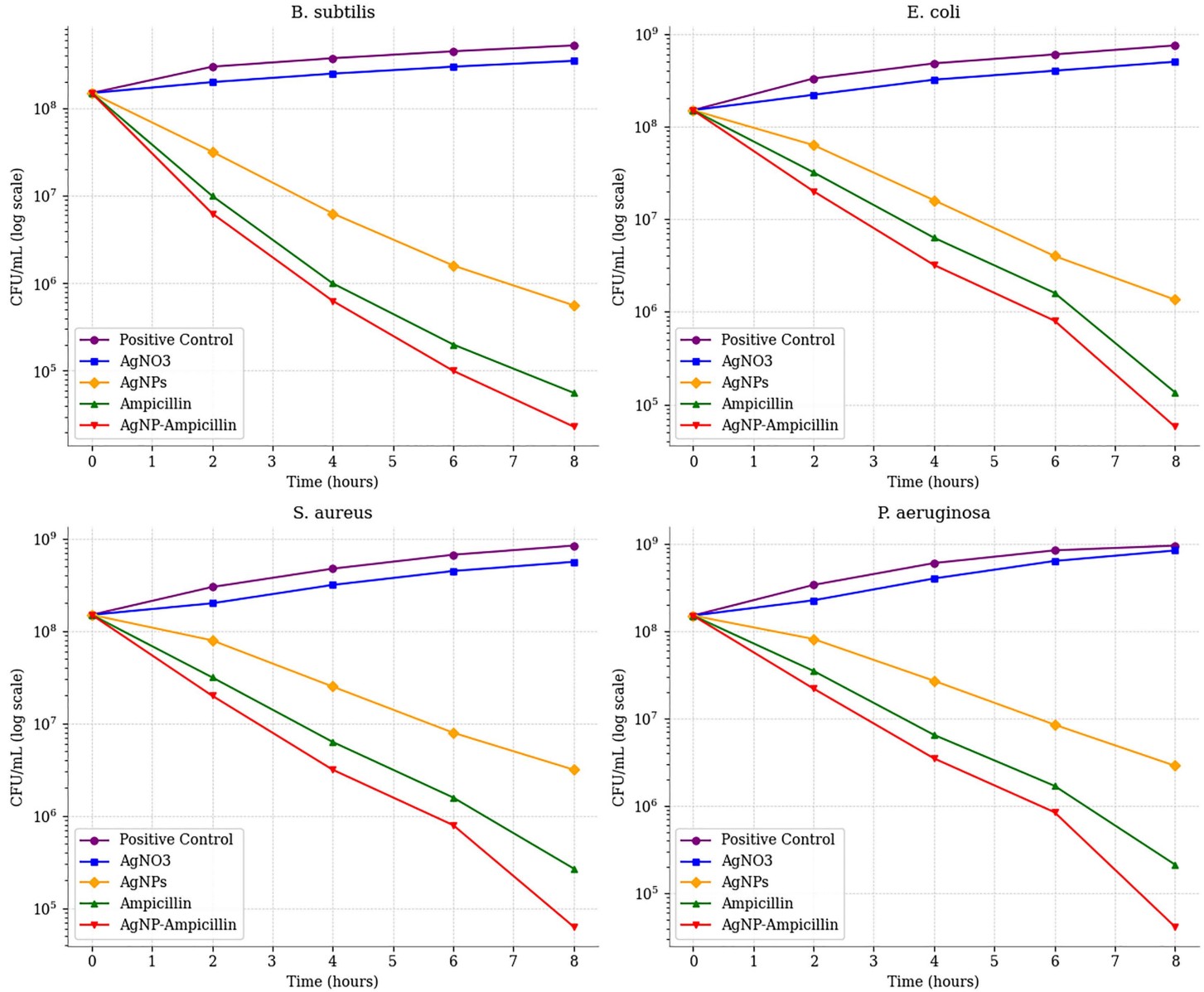

**Fig 9. Time-kill curve of the test samples; (A) AgNO$_3$, (B) AgNPs, (C) Ampicillin, (D) AgNP-ampicillin.**

**Table 10. FIC index of AgNP-ampicillin against resistant clinical isolates.**

| Clinical isolates | MIC (AgNPs alone) | MIC (Ampicillin alone) | MIC (AgNPs in conjugate) | MIC (Ampicillin in conjugate) | FIC Index | Comment |
|---|---|---|---|---|---|---|
| *B. subtilis* | 12.50 µg/mL | 12.50 µg/mL | 1.25 µg/mL | 1.875 µg/mL | 0.25 | Strong synergistic effect |
| *E. coli* | 12.50 µg/mL | 12.50 µg/mL | 1.25 µg/mL | 1.875 µg/mL | 0.25 | Strong synergistic effect |
| *S. aureus* | 12.50 µg/mL | 12.50 µg/mL | 2.50 µg/mL | 3.75 µg/mL | 0.50 | Moderate synergistic effect |
| *P. aeruginosa* | 25.0 µg/mL | 25.0 µg/mL | 5.0 µg/mL | 7.50 µg/mL | 0.50 | Moderate synergistic effect |

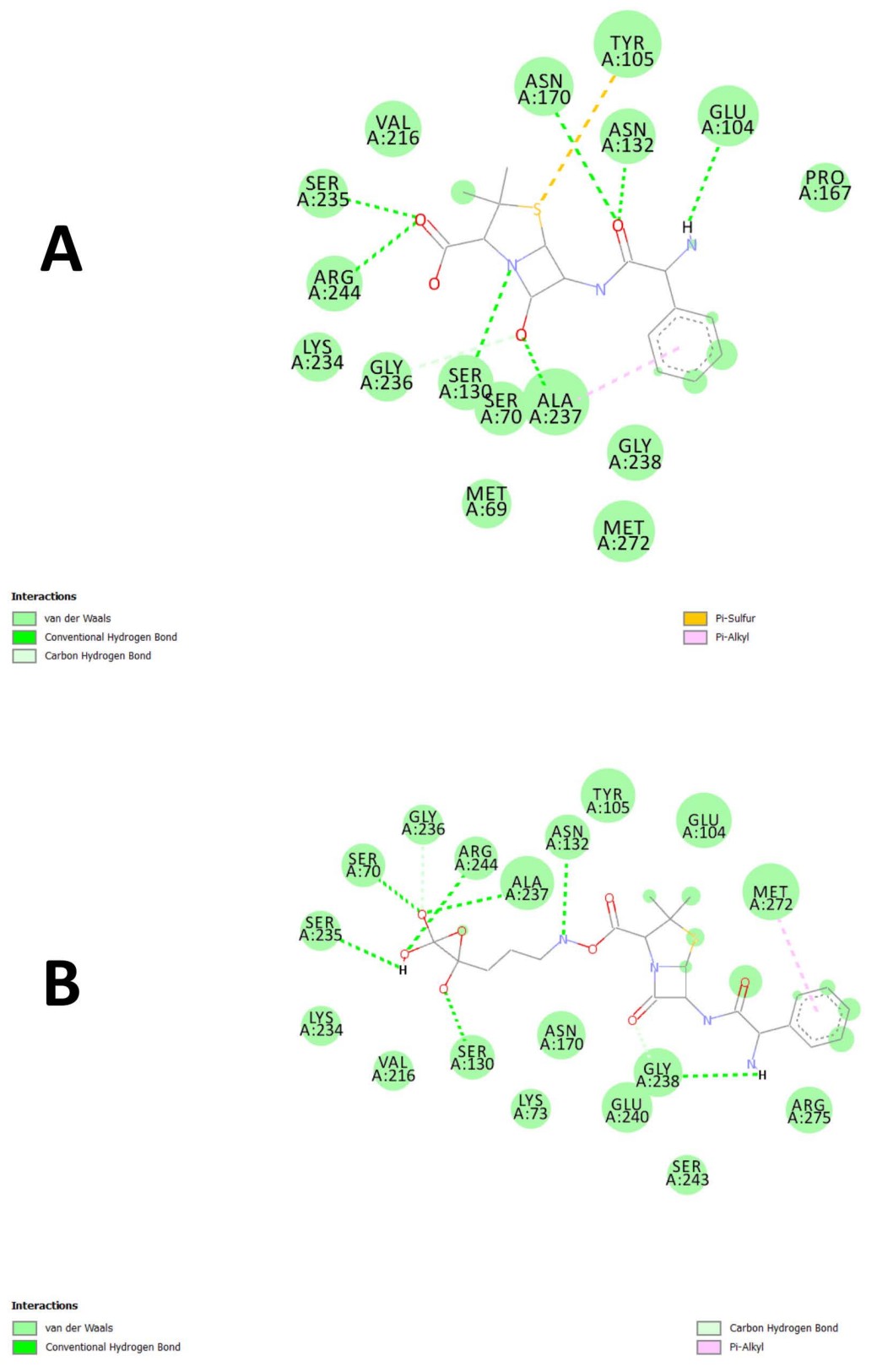

**Interactions**

van der Waals
Conventional Hydrogen Bond
Carbon Hydrogen Bond

Pi-Sulfur
Pi-Alkyl

**Interactions**

van der Waals
Conventional Hydrogen Bond

Carbon Hydrogen Bond
Pi-Alkyl

**Fig 10. Molecular docking binding pose.** (A) Ampicillin. (B) AgNP-ampicillin.

**3.5.3. Structural insight.** Structurally, ampicillin fits better into the active site of Beta-lactamase, forming stronger and more favorable interactions, as evidenced by its higher binding affinity. This phenomenon suggests that ampicillin may have a higher potential for inhibiting Beta-lactamase activity. AgNP-ampicillin, despite its slightly lower binding affinity, may offer distinct advantages due to its conjugated structure, which could facilitate interactions such as π-π stacking or metal coordination with the active site. These findings align with previous research highlighting the importance of structural complementarity and specific interactions in determining ligand-binding affinity.

## 4. Discussions

### 4.1. Relevance of selected bacterial strains

*Bacillus subtilis* is a Gram-positive, rod-shaped bacterium capable of sporulation, which is commonly found in soil and in your gastrointestinal tract. However, its ability to cause opportunistic infections in immunocompromised individuals led it to be primarily used as a model organism for the study of Gram-positive bacterial behavior despite it being non-pathogenic in nature [69].

*Escherichia coli* is a rod-shaped, Gram-negative, facultative anaerobic organism found in the lower intestines of warm-blooded animals Pathogenic strains of *E. coli* are responsible for urinary tract infections, gastrointestinal infections, and sepsis which aids its status as a critical clinical concern, further compounded by its methicillin-resistance [69].

*Staphylococcus aureus* is a biofilm forming Gram positive cocci capable of developing resistance to methicillin (MRSA). MRSA is one of the main causes of nosocomial infection and can lead to multiple diseases, including skin infections, pneumonia, and bloodstream infections. It becomes notably hard to treat due to its resistance to β-lactam antibiotics [69].

*Pseudomonas aeruginosa* is a Gram-negative bacterium shaped like a rod that is often related to hospital-acquired infections in particular patients with weak immune systems. With its efflux pumps, low membrane permeability, and biofilm-forming ability, it has intrinsic resistance to numerous antibiotics and serves as a model organism for the study of drug-resistant infections [69].

### 4.2. Antimicrobial mechanism of AgNP-ampicillin

The AgNP-ampicillin conjugate exhibited markedly more potent antimicrobial action against the tested pathogens than either the pure ampicillin or AgNPs alone, as evidenced by lower values of MIC, MBC, MPC, MBIC, MBEC in addition to rapid bactericidal activity observed in time-kill assay. This increased activity may be attributed to several interconnected mechanisms.

First, a synergistic interaction between AgNPs and ampicillin against resistant pathogens. The bactericidal properties of AgNPs, which include the generation of reactive oxygen species (ROS) and disruption of bacterial membranes, complement the bacteriostatic action of ampicillin [70]. Moreover, AgNPs prevent biofilm formation by modifying the adhesion of bacterial cells, which provides ampicillin with a greater scope to kill isolated bacterial colonies [71].

Second, the nano-scale size of AgNPs enables them to penetrate bacterial cell walls efficiently [28]. This property facilitates the direct transport of conjugated ampicillin molecules to their sites of action within the bacteria, thereby accelerating the antibacterial effect and overcoming certain forms of bacterial resistance. Besides, AgNPs interact with bacterial proteins and DNA, rapidly disrupting essential metabolic processes and assist in the ampicillin-mediated inhibition of peptidoglycan synthesis [71].

Third, conjugation with AgNPs can protect the β-lactam ring of ampicillin from degradation by β-lactamase enzymes, which are commonly produced by resistant bacteria [72]. This protection prolongs the active lifespan of ampicillin, enabling it to exert its bactericidal effects for a longer duration and increasing its effectiveness against resistant strains.

Finally, the conjugated product could act as a prodrug, undergoing hydrolysis to yield four ampicillin molecules at the infection site, thereby improving the local concentration of the antibiotic, particularly for wound or localized infections. Subsequently, the functionalized AgNP would be taken up by liver cells, transported into bile ducts, enter the intestines, and then be eliminated from the body through feces. [73,74]. In animal models, such conjugates reduced bacterial counts at infection sites more effectively than free drugs and promoted faster tissue recovery [75]. Previous studies also

supported the fact that antibiotic-loaded nanoparticles provide more sustained and targeted drug release, resulting in enhanced bacterial killing and improved wound healing compared to free antibiotics [76].

### 4.3. Connection to existing literature

This study presents encouraging developments toward nanoparticle-based antimicrobial therapies against multidrug-resistant pathogens. Compared with the literature already published, the results of this work present some advantages and improvements and give new access to the synthesis, characterization, and biological action of AgNP-ampicillin.

The synergistic antibacterial activity of AgNP-ampicillin, which was reported by Rogowska *et al.* (2017) and Khatoon *et al.* (2019), showed that AgNP-ampicillin has increased bactericidal effects against resistant strains compared to ampicillin or nanoparticles alone [30,28]. Nevertheless, the conjugation efficiency was not precisely quantified; therefore, the mechanisms by which the conjugates interact with the bacterial cells could not be fully elucidated. In addition to quantification of conjugation efficiency, which shows around four molecules of ampicillin were conjugated per AgNP with an effective binding rate of 57.7%, the current study fills these gaps by optimizing synthesis steps such as silica coating, amine functionalization, and carbodiimide-mediated conjugation.

The synergistic bacteriostatic activity of AgNP-ampicillin conjugates is aligned with previous findings, although the current study provides a broader assessment of their effectiveness. While previous reports showed lower MIC for the same class of conjugates, this study determined various other antibacterial endpoints such as MIC, MBC, MPC, MBIC, and MBEC. MIC and MBC values of AgNP-ampicillin conjugates. These values were significantly lower in AgNP-ampicillin conjugates than ampicillin or AgNPs alone, like results found in the investigation of Brown *et al.* (2012) and de Oliveira *et al.* (2017) [27,29]. Besides, similar studies are available for these conjugates, where only the planktonic growth was tested. However, this assay-based study of biofilms presented in this research demonstrates how these compounds act better in biofilm inhibition and biofilm eradication. The finding is crucial since biofilm formation is a significant resistance mechanism of pathogens that must be addressed.

Another significant advancement over previous studies is the β-lactamase protection of ampicillin, which was also demonstrated in this work through molecular docking analysis. Khatoon *et al.* (2019) highlighted the ability of AgNPs to improve the effectiveness of antibiotics but not the methods by which this protection is afforded [28]. The present study reveals that ampicillin conjugated to AgNPs prevents the β-lactam ring from hydrolysis and prolongs the antibiotic activity against resistant strains. Mechanistically, the AgNP-ampicillin conjugates proposed their antimicrobial activity from the synergistic bactericidal effects of AgNPs and ampicillin's bacteriostatic action. The outcome of this study supports the previous work by acknowledging the contribution of potential synergistic effects in antibiotic-nanoparticle combinations.

Moreover, the study reveals important information about the cytotoxicity of AgNP-ampicillin conjugates, which showed >95% cell viability in Vero cells. Because the direct interaction between nanoparticles and mammalian cells is minimized by the "shielding effect" of ampicillin, this finding overcomes the drawbacks of the high cytotoxicity of standalone AgNPs reported by Calderón-Jiménez *et al.*, 2017 [19]. The improved biocompatibility opens up the possibility of clinical applications for the conjugates.

To recapitulate, this study's findings address gaps in conjugation efficiency quantification, synthesis reproducibility, and biofilm-related efficacy. The study establishes AgNP-ampicillin conjugates as a biofilm-inhibiting platform for commonly found multidrug-resistant infections.

### 4.4. Limitations

This study demonstrates the efficacy of AgNP-ampicillin conjugates in vitro against ampicillin-resistant clinical isolates. However, further studies are required to validate these findings in vivo, which is essential for establishing their efficacy in complex biological environments and their pharmacokinetics and pharmacodynamics profiles.

Additionally, long-term stability testing of the conjugates under physiological and storage conditions has not been conducted. Such studies would provide critical insights into their shelf life and clinical applicability.

Besides, while the molecular docking results provide a hypothesis for β-lactamase protection, further experimental validation through enzyme inhibition assays is required to substantiate this mechanism.

## 5. Conclusion

The increase in bacterial resistance has led to the shift towards using multiple broad-spectrum antibiotics to eliminate resistant strains. In this study, the number of ampicillin molecules coupled to each AgNP was quantified, where four ampicillin molecules were determined to be conjugated to each AgNP molecule (Table 4). Increasing the ampicillin molecules conjugated to each AgNP would reduce the required amount of AgNPs for conjugation and, subsequently, the inherent toxicity of AgNPs, while increasing the bioavailability of ampicillin. This reduction in toxicity might be due to the shielding effect of conjugated ampicillin, which minimizes direct nanoparticle interactions with cells.

Additionally, the mechanism of synthesis of AgNP-ampicillin conjugates was elucidated, demonstrating successful functionalization with amine groups on well-defined AgNP surfaces via the use of APTES, followed by conjugation with the carboxyl groups of the ampicillin to form stable amide bonds. The AgNP-ampicillin conjugates showed a considerably greater bactericidal activity than pure ampicillin and AgNPs, which could be attributed to synergistic effects. In particular, the conjugates exhibited greater penetration of bacterial cell walls due to the small size of the AgNPs, and less inactivation of ampicillin by β-lactamases since the β-lactam ring is shielded from enzyme degradation by the conjugation.

In conclusion, the study reveals AgNP-ampicillin conjugates as an appealing platform for combating the resistance of pathogenic bacteria. These conjugates, which combine silver nanoparticles' antibacterial activity with antibiotics' specific action, provide a basis for developing next-generation antimicrobial agents. These results encourage further research to investigate their in vivo efficacy, pharmacokinetics, and long-term stability to define their therapeutic potential completely.

## Supporting information

**S1 File.** S1 Figure. Standard calibration curve of pure ampicillin in distilled water at 216 nm. S1 Appendix. UV-visible Spectroscopy Data. S2 Appendix. FTIR Data. S3 Appendix. DLS and Zeta Potential Data. S4 Appendix. SEM Data. S5 Appendix. EDX Data. S6 Appendix. TGA Data. S7 Appendix. AgNP-ampicillin Synthesis Reaction. S8 Appendix. Microbiological Study Data. S9 Appendix. Molecular Docking Data. S10 Appendix. Cytotoxicity Assay Procedure.
(ZIP)

## Acknowledgments

The authors would like to sincerely thank the Researchers Supporting Project Number (RSP2025R301), King Saud University, Riyadh, Saudi Arabia. The authors are also thankful to the Department of Pharmaceutical Chemistry, Faculty of Pharmacy, University of Dhaka, Bangladesh; Centre for Advance Research in Sciences (CARS), University of Dhaka, Bangladesh; and Department of Nanomaterials and Ceramic Engineering (NCE), Bangladesh University of Engineering and Technology (BUET), Bangladesh for providing laboratory support.

## Author contributions

**Conceptualization:** Muhammad Delwar Hussain, Md. Abdul Mazid.

**Data curation:** Muhammad Salehuddin Ayubee, Farhana Akter.

**Formal analysis:** Muhammad Salehuddin Ayubee, Farhana Akter, Nadia Tasnim Ahmed.

**Funding acquisition:** Abul Kalam Lutful Kabir, Mohsin Kazi, Md. Abdul Mazid.

**Investigation:** Muhammad Salehuddin Ayubee, Farhana Akter, Nadia Tasnim Ahmed.

**Methodology:** Muhammad Salehuddin Ayubee, Nadia Tasnim Ahmed.

**Project administration:** Abul Kalam Lutful Kabir, M. Mahboob Hossain, Md. Abdul Mazid.

**Resources:** Abul Kalam Lutful Kabir, Md. Mahboob Hossain, Md. Abdul Mazid.

**Supervision:** Mohsin Kazi, Md. Abdul Mazid.

**Validation:** Md. Mahboob Hossain, Muhammad Delwar Hussain, Mohsin Kazi.

**Visualization:** Muhammad Salehuddin Ayubee.

**Writing – original draft:** Muhammad Salehuddin Ayubee, Farhana Akter.

**Writing – review & editing:** Nadia Tasnim Ahmed, Abul Kalam Lutful Kabir, Md. Mahboob Hossain, Muhammad Delwar Hussain, Mohsin Kazi, Md. Abdul Mazid.

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
