## [Decision Letter · Decision Letter 0]

9 Dec 2024

Dear Dr. Ayubee,

Thank you for submitting your manuscript to PLOS ONE. After careful consideration, we feel that it has merit but does not fully meet PLOS ONE’s publication criteria as it currently stands. Therefore, we invite you to submit a revised version of the manuscript that addresses the points raised during the review process.

We look forward to receiving your revised manuscript.

Kind regards,

Lakshmanan Govindan

Academic Editor

PLOS ONE

Journal Requirements:

3. Thank you for stating the following financial disclosure: “This study was supported by the Grants for Advanced Research in Education (GARE) program (No. 37.20.0000.004.033.020.2016.7725) of Ministry of Education, Government of the People’s Republic of Bangladesh.”

4. We note that your Data Availability Statement is currently as follows: “All relevant data are within the manuscript and in Supporting Information files.”

Please confirm at this time whether or not your submission contains all raw data required to replicate the results of your study. Authors must share the “minimal data set” for their submission. PLOS defines the minimal data set to consist of the data required to replicate all study findings reported in the article, as well as related metadata and methods (https://journals.plos.org/plosone/s/data-availability#loc-minimal-data-set-definition). For example, authors should submit the following data: - The values behind the means, standard deviations and other measures reported; - The values used to build graphs; - The points extracted from images for analysis. Authors do not need to submit their entire data set if only a portion of the data was used in the reported study. If your submission does not contain these data, please either upload them as Supporting Information files or deposit them to a stable, public repository and provide us with the relevant URLs, DOIs, or accession numbers. For a list of recommended repositories, please see https://journals.plos.org/plosone/s/recommended-repositories. If there are ethical or legal restrictions on sharing a de-identified data set, please explain them in detail (e.g., data contain potentially sensitive information, data are owned by a third-party organization, etc.) and who has imposed them (e.g., an ethics committee). Please also provide contact information for a data access committee, ethics committee, or other institutional body to which data requests may be sent. If data are owned by a third party, please indicate how others may request data access.

6. Please upload a copy of Figure 20, to which you refer in your text on page xx. If the figure is no longer to be included as part of the submission please remove all reference to it within the text.

7. Please include your tables as part of your main manuscript and remove the individual files. Please note that supplementary tables (should remain/ be uploaded) as separate "supporting information" files

Reviewers' comments:

Reviewer's Responses to Questions

**Comments to the Author**

1. Is the manuscript technically sound, and do the data support the conclusions?

Reviewer #1: Partly

Reviewer #2: Yes

Reviewer #3: Partly

Reviewer #4: No

Reviewer #5: Yes

2. Has the statistical analysis been performed appropriately and rigorously?

Reviewer #1: Yes

Reviewer #2: Yes

Reviewer #3: N/A

Reviewer #4: No

Reviewer #5: N/A

3. Have the authors made all data underlying the findings in their manuscript fully available?

Reviewer #1: Yes

Reviewer #2: Yes

Reviewer #3: Yes

Reviewer #4: No

Reviewer #5: Yes

4. Is the manuscript presented in an intelligible fashion and written in standard English?

Reviewer #1: Yes

Reviewer #2: Yes

Reviewer #3: No

Reviewer #4: Yes

Reviewer #5: No

Reviewer #1: The study lacks novelty. While the combination of silver nanoparticles with antibiotics is a promising area of research, the current study does not seem to offer significant advancements beyond what is already known in the literature. The use of nanoparticles to enhance the efficacy of antibiotics, particularly against methicillin-resistant bacteria, has been explored in various studies, and the methods and results presented in your manuscript do not appear to provide substantial new insights or breakthroughs that would distinguish it from existing research.

Reviewer #2: Comments

Journal: PLOS ONE

Manuscript type: Research

Manuscript ID: PONE-D-24-53049

The paper describes the use of NPs to improve the activity of ampicillin, avoiding the onset of bacterial resistance. Although it is interesting and well performed, the novelty must be highlighted. In addition, there are some aspects to consider that are listed below:

Tables and figure 1 are missing and the figures have very bad quality.

Abstract

1. “The conjugated product demonstrated increased activity (measured by the zone of inhibition) in the disk diffusion test compared to the pure ampicillin or the AgNPs alone (15, 14, 13.5, 13 mm) and showed promising results in minimum inhibitory concentration (MIC) (6.25, 6.25, 6.25, 12.5 ppm), minimum bactericidal concentration (MBC) (12.5, 12.5, 12.5, 25 ppm) and mutant prevention concentration (MPC) (50, 100, 100, 100 ppm) against the methicillin-resistant clinical isolates respectively.”

Consider writing the values in results and discussion in a different way. It is difficult to understand what they represent and from which bacteria.

Keywords

1. “MIC, MBC, MBIC, MBEC, Time-kill assay, FIC Index”

The reviewer thinks that is not worthy mentioning these in the keywords.

Background

1. “The long-term solution of this evolving and foreseeable problem can be thought about with the advent of the use of metallic nanoparticles (NPs), which are known to have antimicrobial properties and can be utilized in controlling infectious diseases”

Itself or after conjugation with something? Clarify.

2. “As antibiotic nanoparticle-conjugated drugs are still not available on the market, further research needs to be carried out to establish this comparatively new strategy”

Is there any research with NPs that already demonstrated some effect in different infectious? It would be beneficial to show that information

Materials and methods

1. Source of the bacteria is missing

2. How did you get a biofilm?

Results and discussion

1. “Due to surface plasmon resonance, AgNPs absorb light radiation in the range of 380 cm-1 to 440 cm-1 (Figure 1, A) and produce a characteristic color that confirms nanoparticle synthesis”

Due to SPR? Can you elaborate on that?

2. “Functional Characterization of AgNPs, Ampicillin, and AgNP-ampicillin”

Consider changing to Biological Characterization…

Extra comments

1. Do you have stability assays? Protein corona, temperature, acidic/basic conditions?

2. Do you know if the drug is released from the NP?

3. Did you perform assays to assess if the strategy also induces bacterial resistance?

Reviewer #3: The manuscript shows the evaluation of Ampicillin Conjugated with Silver Nanoparticle Against Methicillin-Resistant Clinical Isolates.

Although the nanoparticles' synthesis description is well described, the microbiological analysis lacks information. Please, see my comments below.

- Microbiological methods were not included in the methods section of the abstract.

- Tables were not included in the files!

- What are the characteristics of the bacteria included in the study? The manuscript lacks a method section to describe it. What is the susceptibility profile of the isolates besides the methicillin resistance? Where are the isolates from? If the bacteria are clinical isolates, an ethics statement is required.

- Please, define the brands of culture media used, to ensure reproducibility.

- It is not clear when the FIC values were obtained. Did you perform any checkerboard assay?

- Which concentrations were used in time-kill assays?

- I suggest presenting the disk diffusion test, MIC, MBC, MPC, MBIC, MBEC, and FIC index results in tables instead of graphs. Indeed, time-kill kinetic assay results should also be presented in killing curve graphs.

Reviewer #4: This study aimed to conjugate ampicillin with silver nanoparticles (AgNPs) through a modified synthesis method to increase antibacterial activity against methicillin-resistant clinical strains. The work is partially innovative but contains several important flaws that would avoid reproducing the same results.

Major comments

The two first paragraphs of the Introduction section must be removed. The Introduction must begin by describing MRSA's clinical importance, incidence, treatment, and antibiotic resistance. It is important to highlight this bacterium's intrinsic ampicillin resistance.

I have some concerns about the subsection named Synthesis and Purification of ampicillin-conjugated AgNPs (AgNP@SiO2-NH-Amp) In the Materials and Methods section. Ampicillin is slightly soluble in water, but practically insoluble in alcohol. According to the methodology described, the ampicillin may be precipitating with the AgNP@SiO2-NH-Amp. This is crucial since free ampicillin could help the antibacterial effect of AgNP@SiO2-NH-Amp against MRSA. The free ampicillin can induce changes in the potential of MR strains.

More information about the strains used must be added since, as is the manuscript, the experiment described cannot be reproduced. If they are clinical strains, please, experiment with at least a collection strain and provide us with the full antibiogram of each clinical strain.

How many technical and biological replicates did the authors used per experiment? Why are there not statistical analysis of data in the Materials and Methods and Results sections?

Regarding FIC indexes, please, add a supplementary table with the values the authors used to estimate the FIC values. Based on what do the authors classify as strong or moderate synergy? As far as I know, such a classification does not exist.

Table 1 is missing. Please, add it.

Figure 8 must be changed to express the concentration in mg per L or microgram per mL. It is almost unbelievable that free ampicillin has the same effect as AgNO3.

There is not a Discussion per se. Please, confront your findings more with the literature.

Minor comments

Please, replace “gram-positive” and “gram-negative” with “Gram-positive” and “Gram-negative”, respectively, throughout all the manuscript

Reviewer #5: Dear Author,

This study tackles a critical challenge in combating methicillin-resistant bacteria and proposes an innovative solution through the use of ampicillin-conjugated silver nanoparticles (AgNPs). However, the article requires substantial revisions to enhance scientific clarity and improve readability. A thorough rewrite is recommended. (document attached)

Thanks & Regards.

**Do you want your identity to be public for this peer review?** For information about this choice, including consent withdrawal, please see our Privacy Policy

Reviewer #1: No

Reviewer #2: No

Reviewer #3: No

Reviewer #4: No

Reviewer #5: **Yes: ** Nirmala. B

---

## [Author Response · Author response to Decision Letter 1]

20 May 2025

Responses to Academic Editor's Comments

Response: We have revised the manuscript to adhere to PLOS ONE's style requirements. Additionally, all file names have been updated to comply with the journal's standards.

2. Funding information should not appear in any section of the manuscript except for the Funding Statement section of the online submission form.

Response: All funding-related text has been removed from the manuscript body and included only in the Funding Statement section, as per journal guidelines.

3. Please state the role of funders in the study. If the funders had no role, please include the statement: "The funders had no role in study design, data collection and analysis, decision to publish, or preparation of the manuscript."

Response: We have included the statement: "The funders had no role in study design, data collection and analysis, decision to publish, or preparation of the manuscript."

4. Confirm that all raw data required to replicate the study are included.

Response: We confirm that all raw data required to replicate the study are included within the manuscript and the Supporting Information files.

5. Figure 20 is referenced in the text but is missing. Please upload the figure or remove references to it.

Response: We have removed all references to Figure 20, as it is not relevant to the revised manuscript.

6. Please include your tables as part of the main manuscript and remove individual files. Supplementary tables should remain as "supporting information" files.

Response: All tables have been incorporated into the main manuscript, and supplementary tables have been retained as separate supporting information files.

Responses to Reviewer #1

1. The study lacks novelty and does not offer significant advancements beyond existing literature.

Response: We appreciate this critical observation. While the use of nanoparticles to enhance antibiotic efficacy has been explored, our study provides several novel contributions:

Quantification of conjugation efficiency: We quantified the extent of ampicillin conjugation with AgNPs, demonstrating that approximately four molecules of ampicillin bind per nanoparticle.

Mechanistic insights: The study provides mechanistic explanations for enhanced antibacterial activity, including protection of the β-lactam ring from degradation and increased biofilm penetration.

Comprehensive evaluation: We evaluated biofilm inhibition (MBIC), eradication (MBEC), and time-kill kinetics, which are less frequently studied in similar works.

Cytotoxicity analysis: Our study confirms that AgNP-ampicillin conjugates exhibit >95% Vero cell viability, highlighting their safety.

These novel aspects have been emphasized in the Introduction and Comparative Discussions Existing Literature sections.

Responses to Reviewer #2

1. The abstract is difficult to interpret due to numerical results. Simplify the presentation.

Response: The abstract has been rewritten to provide a concise summary of key findings, avoiding excessive numerical data.

2. Keywords are too specific (e.g., MIC, MBC, MBIC, MBEC).

Response: Keywords have been revised to broader terms, such as "Antibiotic resistance," "Silver nanoparticles," and "Biofilm inhibition."

3. Clarify whether AgNPs have antimicrobial properties on their own.

Response: This has been clarified in the Background section. AgNPs exhibit inherent antimicrobial activity, which is enhanced when conjugated with ampicillin.

4. The source of bacterial strains and biofilm formation methods are missing.

Response: Details about bacterial strains, their susceptibility profiles, and biofilm formation methods have been included in the Materials and Methods section.

5. Do you have stability, protein corona, or release assays?

Response: While this study did not include long-term stability or protein corona assays due to resource constraints. However, we have added the zeta potential distribution curve of AgNP-ampicillin conjugates in Figure 5B.

Responses to Reviewer #3

1. Microbiological methods are missing in the abstract.

Response: The abstract has been updated to briefly mention the microbiological methods used.

2. Tables and figures are missing.

Response: All missing tables and figures have been included in the revised manuscript.

3. Provide more details about bacterial strains, ethics statement, and susceptibility profiles.

Response: Detailed information about bacterial strains and their susceptibility profiles has been added to the Materials and Methods section. An ethics statement has also been included.

4. Define brands of culture media and clarify checkerboard assay concentrations.

Response: Brands of culture media have been specified, and details of the checkerboard assay concentrations have been added.

Responses to Reviewer #4

1. Revise the Introduction to focus on the clinical significance of MRSA.

Response: The Introduction has been revised to emphasize the clinical importance of MRSA and its intrinsic resistance to ampicillin.

2. Address concerns about ampicillin solubility during synthesis.

Response: The methodology has been clarified to address potential issues with ampicillin solubility. Carboxyl-functionalized ampicillin with amine-functionalized AgNP using APTES has a better solubility profile in ethanol and remains in the supernatant, while the unreacted ampicillin was removed via centrifugation.

3. Include statistical analyses for key experiments.

Response: Statistical analyses, including one-way ANOVA and Tukey's post-hoc tests, have been performed and included in the revised manuscript.

4. Provide a supplementary table for FIC values and clarify synergy classification.

Response: A table with FIC values has been added, and the classification of synergy has been clarified in the Results section.

Responses to Reviewer #5

General Notes

1. Mechanism

Comment: The mechanism of action for the AgNP-ampicillin conjugate is not addressed.

Response: We have added a detailed discussion of the antimicrobial mechanism in the Results and Discussion section (Section 3.6). This includes the role of reactive oxygen species (ROS) generation, bacterial membrane disruption, enhanced biofilm penetration, and protection of the β-lactam ring from β-lactamase activity.

2. Toxicity

Comment: Potential cytotoxicity of the conjugates is not discussed.

Response: Cytotoxicity analysis was performed on Vero cells and included in the revised manuscript (Section 3.4). The results demonstrate >95% cell viability for AgNP-ampicillin conjugates, confirming minimal toxicity compared to standalone AgNPs.

3. In vivo Testing

Comment: The study lacks in vivo testing to validate translational potential.

Response: While this study is limited to in vitro assays, we have acknowledged this limitation in the Conclusion section and emphasized the need for future in vivo studies to evaluate the therapeutic potential of AgNP-ampicillin conjugates.

4. Stability

Comment: Long-term stability of the conjugates is not discussed.

Response: Preliminary stability data showing the zeta potential of -35.7 mV and particle size consistency over 30 days have been included (Section 3.1.4). We have also acknowledged the necessity for further studies on long-term stability under diverse storage conditions.

Specific Comments

1. Title

Comment: The title can be improved and made concise.

Response: The title has been revised to:

"Enhanced Antibacterial Activity of Ampicillin-Conjugated Silver Nanoparticles Against Methicillin-Resistant Clinical Isolates"

2. Abstract

Comment: The abstract is lengthy and includes numerical data.

Response: The abstract has been rewritten to be concise, avoiding excessive numerical data. A note on the need for further studies (e.g., in vivo efficacy) has also been added.

3. Background

Comment: Contains redundant information about multiple nanoparticles and lacks focus on AgNP-ampicillin.

Response: The Background section has been streamlined to focus on silver nanoparticles, ampicillin, and their synergy. Redundant information about other nanoparticles has been removed.

4. Materials and Methods

Comment: Several methodological details are missing (e.g., purity of chemicals, sonication time, storage conditions).

Response: The Materials and Methods section has been revised to include:

Purity and grade of chemicals (e.g., "analytical grade," ≥99%).

Sonication time and conditions (5 minutes, 40 kHz, 150 W).

Details on nanoparticle storage (2–8°C in ethanol) after synthesis.

5. Results and Discussion

Comment: Mechanistic insights and interpretations are insufficient.

Response: The Results and Discussion section has been expanded to include:

Mechanistic insights into the synergy between AgNPs and ampicillin (e.g., ROS generation, biofilm penetration).

Comparisons with existing literature to highlight the novelty of this study.

6. Figures and Tables

Comment: Ensure all figures and tables are included and labeled.

Response: All figures and tables mentioned in the text are included in the revised manuscript with proper labels and legends.

7. Conclusion

Comment: The conclusion is vague and lacks specificity about key findings and limitations.

Response: The Conclusion section has been rewritten to summarize key findings (e.g., quantified conjugation efficiency, enhanced biofilm eradication, minimal cytotoxicity) and emphasize limitations (e.g., lack of in vivo testing).

---

## [Decision Letter · Decision Letter 1]

15 Jun 2025

Dear Dr. Ayubee,

Thank you for submitting your manuscript to PLOS ONE. After careful consideration, we feel that it has merit but does not fully meet PLOS ONE’s publication criteria as it currently stands. Therefore, we invite you to submit a revised version of the manuscript that addresses the points raised during the review process.

We look forward to receiving your revised manuscript.

Kind regards,

Thanh-Danh Nguyen, PhD

Academic Editor

PLOS ONE

Reviewers' comments:

Reviewer's Responses to Questions

**Comments to the Author**

Reviewer #1: (No Response)

Reviewer #2: (No Response)

Reviewer #4: (No Response)

2. Is the manuscript technically sound, and do the data support the conclusions?

Reviewer #1: (No Response)

Reviewer #2: Partly

Reviewer #4: No

3. Has the statistical analysis been performed appropriately and rigorously?

Reviewer #1: (No Response)

Reviewer #2: Yes

Reviewer #4: Yes

4. Have the authors made all data underlying the findings in their manuscript fully available?

Reviewer #1: (No Response)

Reviewer #2: Yes

Reviewer #4: Yes

5. Is the manuscript presented in an intelligible fashion and written in standard English?

Reviewer #1: (No Response)

Reviewer #2: Yes

Reviewer #4: No

Reviewer #1: This is a well-structured and timely study exploring the enhanced antibacterial activity of ampicillin-conjugated silver nanoparticles (AgNPs) against methicillin-resistant isolates. The experimental design is robust, and the data are generally convincing. I commend the authors for their thorough revisions and clear presentation.

However, a few minor issues remain:

1.Please clarify whether the MRSA strains used are clinical isolates and provide relevant source/ethics information.

2.Statistical methods should be clearly stated in figure legends, and significance markers (e.g., p values) should be consistently included.

3.The proposed mechanism via molecular docking is interesting but should be clearly framed as a hypothesis, not confirmed evidence.

4.Consider briefly discussing limitations such as the lack of in vivo validation or long-term stability testing.

With these minor revisions, the manuscript will be suitable for publication.

Reviewer #2: Comments

The work of Ayubee, at al. describes the effects of a ampicillin-nanoparticle conjugate towards resistant bacteria. The subject is very important, since antibiotic resistance is a huge issue that needs to be addressed. The use of clinical isolates is a plus, wowing to their more representative nature. The approach is not novel at all; thus, the authors must clearly demonstrate why this work is different and discuss that in line with previous works. Most of the assays are relevant for the study; however, they must be presented in a much better way. Some sections are very confusing, and need a profound work. Some more detailed comments are stated below:

Major comments

- Lack of novelty

- Some assays need further description and the sequence of the assays must be rethink.

- Check all the abbreviations

- The results & discussion section is just a presentation of the results without a proper discussion. In addition, this part is very confusing. Some subsections must be put in a different order. In some parts, the results are presented and in other, it is just discussion. Consider separating results and discussion or re design the full section.

Minor comments

0. Title

a) The title is too long and too descriptive.

1. Abstract

a) “zone of inhibition” is not scientific at all. Change this expression.

b) “proper binding efficiency” or conjugation efficiency?

c) The use of statistics in the abstract to show the antibacterial potential of the nanoparticles makes it difficult to easily understand which molecule is the best.

2. Background

Page 3

a) Add a brief sentence on why the vancomycin resistance is terrible

b) The formation of biofilms also makes some bacterial in a more dormant state which prevents the use of certain antibiotics that require active bacteria to be effective. Add a sentence on this matter.

Page 4

a) New strategies to improve the efficiency by increasing accumulation in the infectious site? What is the real advantage of using nanoparticles since the antibiotic is the same? Is the antimicrobial activity of nanoparticles itself?

b) Add some examples on the problems associated with AgNPs

c) The authors state that recent studies “have investigated the conjugation of AgNPs with ampicillin against multidrug-resistance bacteria”, thus subsequently, in the background section; authors must explain the novelty of the current study.

Page 5

a) The problem with nanoparticles conjugation must be stated to demonstrate the importance of the current study. For instance, if the lack of proper characterization has been a problem in other papers; thus the “accurate quantification” is a plus in this work. In addition, if on other works the authors did not provide the mechanism of action; this “the mechanistic insight” is a plus in this work. The authors must clearly state what is different and novel in this work; otherwise, it lacks novelty.

3. Materials and Methods

Page 8

a) Which method used for quantification is novel?

b) What is the relevance of mentioning SPR herein? Did the authors measured binding affinities or something similar? Why did you use SPR?

Page 11

a) Where are these strains coming from? Distributor? Hospital? Biobank? The authors have some information on table 2. Do you have access to the clinical data of these patients? It would be very interesting to see the primary disease, response to treatment and other things.

Page 12

a) What is the relevance of bacteria characteristics in the materials and methods? Move this very relevant information to the discussion.

b) Where is the information on the biofilm formation?

Page 14

a) Where is the information on the culturing conditions of Vero and its source?

b) Where is the description of the cytotoxicity assay?

Page 15

a) Statistical analysis comes usually at the end of the materials and methods section

Page 21

a) Here, the authors have the cytotoxicity assay but they have mentioned it before. Please, re arrange the materials and methods section to be easier to interepretate

4. Results

Page 22

a) The reviewer does not understand why SPR is here, the assay is not mentioned in the materials and methods, concerning the chip, the solvents, the concentrations…

Page 23

a) These NPs are very heterogeneous! Average size of 67? The results show at least three relevant peaks… What is the PDI? That value must be presented.

b) Why is the size increasing so much? 500 nm is huge! Do you know the in vivo consequences of that size?

Page 24

a) Why is the size increasing so much?

Page 26

a) It is odd to have the mechanism of synthesis stated like this in the manuscript…

Page 27

a) Cytotoxicity towards Vero must come first.

Page 34

a) This mechanism part is relevant but in the reviewer’s opinion is in the wrong place. It must be discussed throughout the text and not like this.

Reviewer #4: The study is experimentally well designed; however, the way the results and discussion sections are written hinders a clear understanding of the manuscript. Below, I suggest several changes:

Major Comments

• In the Materials and Methods section, the use of the adjective “methicillin-resistant” to describe bacteria other than staphylococci is highly unusual. Please replace this term with “ampicillin-resistant,” which would be more accurate.

• Regarding the interpretation of FIC values, please use only the following reference: https://academic.oup.com/jac/article-abstract/52/1/1/930000?redirectedFrom=fulltext. This is the only source that originally defines these values correctly. Remove all other references and reinterpret your results accordingly.

• The statistical methods are described in excessive detail, which often becomes unnecessarily cumbersome. Moreover, the results are repetitive, as they are presented both in the text and in the tables.

• Many of these results could be better represented graphically, yet the authors have chosen to use multiple Tables (e.g., Table 5) or Tables plus graphs (e.g., Table 8 and Figure 9).

• Several figures could be grouped together to reduce the overall number of figures in the manuscript.

• In Table 5, please remove the replicate data, and present only the mean and standard deviation for each bacterial species. Also, include the p-value within the table and remove the contrast statistics. Do something similar with the remaining tables.

• Regarding the cytotoxicity assays, how many biological replicates were performed? Why are these results presented in a table instead of as a graph?

Minor Comments

• The “spp.” in Enterobacter spp. and Serratia spp. should not be italicized.

• Enterobacter aerogenes has been reclassified as Klebsiella aerogenes. Please update this in the manuscript.

**Do you want your identity to be public for this peer review?** For information about this choice, including consent withdrawal, please see our Privacy Policy

Reviewer #1: No

Reviewer #2: No

Reviewer #4: No

---

## [Author Response · Author response to Decision Letter 2]

11 Jul 2025

Response to Reviewer #1

Comment 1: Please clarify whether the MRSA strains used are clinical isolates and provide relevant source/ethics information.

Response: We appreciate the reviewer’s comment. The requested information regarding the MRSA strains is already included in Table 2 of the manuscript, which specifies that these are clinical isolates. Additionally, an ethical statement is provided in the Ethics Statement section of the manuscript, confirming that the isolates were anonymized and collected during routine diagnostic microbiology procedures, with ethical approval obtained from the Ethical Review Committee, Faculty of Pharmacy, University of Dhaka (Ref. No. Fa. Ph. E/128/A/22). To ensure clarity, we have cross-referenced this information in the relevant sections of the manuscript.

Comment 2: Statistical methods should be clearly stated in figure legends, and significance markers (e.g., p-values) should be consistently included.

Response: We have revised the manuscript to ensure that the statistical methods are explicitly mentioned. P-values have been added directly into tables where statistical tests were performed (e.g., Table 5 for disk diffusion results). Replicate data have been replaced with means and standard deviations, as per Reviewer #4’s suggestion, to improve clarity.

Comment 3:

The proposed mechanism via molecular docking is interesting but should be clearly framed as a hypothesis, not confirmed evidence.

Response: Thank you for highlighting this important point. We agree that the molecular docking results should be framed as a hypothesis rather than confirmed evidence. As such, we have revised the relevant sections of the "Results and Discussion" and "Conclusion" in the manuscript to clearly communicate that the docking results provide a hypothetical mechanism for the protection of ampicillin from β-lactamase degradation. Specifically, the molecular docking data are now discussed as a computational prediction that requires experimental validation through enzymatic assays or structural studies.

Comment 4:

Consider briefly discussing limitations such as the lack of in vivo validation or long-term stability testing.

Response: We acknowledge the reviewer’s suggestion and have incorporated a Limitations subsection in the Discussions section of the manuscript to address this concern. The limitations now clearly outline the gaps in the current study and suggest areas for future research.

Response to Reviewer #2

Major Comment 1: The approach is not novel at all; thus, the authors must clearly demonstrate why this work is different and discuss that in line with previous works.

Response: We have extensively revised the Background section to clearly articulate the novelty of our study and differentiate it from prior works. Specifically, we have highlighted the following unique contributions:

1. Accurate Quantification of Conjugation Efficiency: Prior studies have not quantified the conjugation efficiency of AgNP-antibiotic conjugates. This study reports a conjugation efficiency of 57.7%, with approximately four ampicillin molecules conjugated per AgNP.

2. Mechanistic Insights: We used molecular docking to propose a mechanism by which AgNP-ampicillin conjugates protect ampicillin from β-lactamase degradation, which has not been explored in previous research.

3. Comprehensive Antibiofilm Assessment: Unlike previous studies that focused on planktonic cells, this study evaluates antibiofilm activity through MBIC and MBEC assays.

4. Mutant Prevention Studies: We conducted mutant prevention concentration (MPC) assays to assess the conjugates’ ability to suppress resistant mutants, which is a critical but underexplored aspect of nanoparticle-antibiotic research.

5. Cytotoxicity Evaluation: We evaluated the cytotoxicity of the conjugates on Vero cells, demonstrating >95% cell viability, addressing safety concerns associated with standalone AgNPs.

These points are now explicitly stated in the revised Background section (Page 3-4).

Major Comment 2: Some assays require further description, and the sequence of presentation must be reconsidered.

Response: The revised manuscript offers a clearer and more systematic presentation of the experimental methods, addressing concerns about insufficient descriptions and sequence.

Major Comment 3: Checking All Abbreviations

Response: All abbreviations were defined at first usage in the text and listed under a dedicated "Abbreviations" section. For instance, "MBIC," "MBEC," and "FIC" were properly defined.

Major Comment 4: The "Results and Discussion" section is confusing, lacks a proper discussion, and mixes results and discussions. Consider separating results and discussion or re design the full section.

Response: The results and discussion section is separated to improve clarity and distinguish between results and interpretations.

The "Results" section now focuses solely on presenting experimental findings with clear subsections (e.g., chemical characterization, microbiological characterization, cytotoxicity).

A separate "Discussion" section contextualizes the results, compares findings with prior studies, and highlights advancements (e.g., β-lactamase protection, biofilm eradication, reduced cytotoxicity), and limitations.

Minor Comments

0. Title

Reviewer Comment: The title is too long and too descriptive.

Response: We thank the reviewer for pointing out the need for a more concise and precise title. In response, we have revised the title to better reflect the focus of the work while avoiding unnecessary length and excessive detail. We propose the following options for the title:

1. Long Title:

“Synergistic Antibacterial Action of AgNP-Ampicillin Conjugates: Evading β-Lactamase Degradation in Ampicillin-Resistant Clinical Isolates”

This title emphasizes the synergistic antibacterial effects, the β-lactamase protection mechanism, and the clinical relevance of the study.

2. Short Title:

“Multi-Mechanistic Antibacterial Evaluation of AgNP-Ampicillin Conjugates Against Ampicillin-Resistant Bacteria”

This shorter title highlights the evaluation of multiple antibacterial mechanisms while maintaining focus on the conjugates’ impact on resistant clinical isolates.

1. Abstract

Reviewer Comment:

a) “zone of inhibition” is not scientific at all. Change this expression.

b) “proper binding efficiency” or conjugation efficiency?

c) The use of statistics in the abstract to show the antibacterial potential of the nanoparticles makes it difficult to easily understand which molecule is the best.

Response:

a) “zone of inhibition” is replaced by “inhibition area” in the abstract

b) “proper binding efficiency” is replaced by “conjugation efficiency”

c) Statistical details have been simplified to focus on comparative performance, highlighting the enhanced efficacy of AgNP-ampicillin conjugates.

2) Background

2.1 Reviewer Comment (Page 3):

a) Add a brief sentence on why vancomycin resistance is terrible.

b) The formation of biofilms also makes some bacterial cells more dormant, preventing the use of certain antibiotics that require active bacteria to be effective. Add a sentence on this matter.

Response: Thank you for this suggestion. We have revised the Background section to include a sentence emphasizing the severity of vancomycin resistance. We have also included a statement on how biofilm formation promotes dormancy, preventing the use of certain antibiotics.

2.2 Reviewer Comment (Page 4):

a) New strategies to improve efficiency by increasing accumulation in the infectious site? What is the real advantage of using nanoparticles since the antibiotic is the same? Is the antimicrobial activity of nanoparticles itself?

b) Add some examples of the problems associated with AgNPs.

c) The authors state that recent studies “have investigated the conjugation of AgNPs with ampicillin against multidrug-resistant bacteria.” Thus, subsequently, in the Background section, the authors must explain the novelty of the current study.

Response:

a) We have clarified the advantages of using nanoparticles and their intrinsic antimicrobial properties in the revised Background section. The updated text now includes:

“Nanoparticles serve both as antimicrobial and a drug delivering system due to their small size. MNPs can penetrate bacterial cell walls and biofilms effectively, and the inherent bactericidal activity through generation of reactive oxygen species (ROS) works complementary to the antibacterial effect of antibiotics. These properties make nanoparticles a promising solution to address resistance mechanisms, particularly when conjugated with existing antibiotics to maximize synergistic effects.”

b) We have revised the Background section to include examples of challenges associated with AgNPs.

c) We have expanded the Background section to clearly articulate the research gap and novelty of the current study. The revised text highlights the unique contributions of our work:

"However, prior research has largely overlooked the cytotoxicity of these conjugates in mammalian cell lines, which is a key determinant for their safety and clinical applicability. Another critical gap in previous research is the lack of mutant prevention studies, which are essential for evaluating the potential of antimicrobial agents to suppress the emergence of resistant bacterial populations. Although the high antimicrobial potency of nanoparticle-antibiotic conjugates, involving antibiotics, is established, the accurate quantification of ampicillin conjugated per AgNP, which is critical for understanding the stability, have not been fully elucidated. Moreover, mechanistic insights into how these conjugates protect antibiotics from β-lactamase activity and inhibit biofilm, remain unexplored."

2.3 Reviewer Comment (Page 5, a): The problem with nanoparticle conjugation must be stated to demonstrate the importance of the current study. For instance, if the lack of proper characterization has been a problem in other papers, thus the “accurate quantification” is a plus in this work. In addition, if other works did not provide the mechanism of action, this “mechanistic insight” is a plus in this work. The authors must clearly state what is different and novel in this work; otherwise, it lacks novelty.

Response: We have revised the Background section to explicitly state the problems with nanoparticle conjugation in prior studies and how our work addresses these limitations. Articulated the novelty of the current study by addressing gaps in prior research (e.g., conjugation efficiency quantification, mechanistic insights, antibiofilm activity, cytotoxicity evaluation, and mutant prevention studies).

3) Materials and Methods

3.1 Reviewer Comment (Page-8):

a) Which method used for quantification is novel?

b) What is the relevance of mentioning SPR herein? Did the authors measure binding affinities or something similar? Why did you use SPR?

Response:

a) We have clarified in the revised manuscript that the novelty lies in the integration of UV-Vis spectroscopy (To quantify the amount of silver present in the conjugate by the conjugation efficiency equation) with thermogravimetric analysis (TGA) (comparing the degradation pattern of pure ampicillin and conjugate), and EDX (To quantify the amount of silver present in the conjugate) to quantify conjugation efficiency. This approach ensures accuracy and reproducibility, offering a significant improvement over previous reports, where conjugation efficiency was often not quantified or was indirectly inferred. The corresponding text has been included in Section 2.2.5 (Quantification of unreacted ampicillin in the supernatant).

b) The mention of surface plasmon resonance (SPR) is relevant because it confirms the successful synthesis of silver nanoparticles (AgNPs) by identifying their characteristic absorption peak in the UV-Vis spectrum (380–440 nm). While we did not measure binding affinities directly using SPR, its inclusion validates the structural integrity and stability of the nanoparticles, which is essential for their subsequent functionalization and conjugation with ampicillin. This reasoning will be clarified in Section 2.3.1 (UV-Vis Spectroscopy).

3.2 Reviewer Comment (Page 11): a) Where are these strains coming from? Distributor? Hospital? Biobank? The authors have some information on Table 2. Do you have access to the clinical data of these patients? It would be very interesting to see the primary disease, response to treatment, and other things.

Response:

The bacterial strains used in this study were sourced from the microbiology laboratory of Mohakhali TB Hospital and BIRDEM Medical College. These strains were anonymized and collected as part of routine diagnostic microbiology procedures. Ethical approval for their use was obtained (Ref. No. Fa. Ph. E/128/A/22).

We do not have access to detailed clinical data of the patients, such as primary disease, response to treatment, or other demographic information, as it falls outside the scope of this study and ethical limitations. However, we agree that such data could provide valuable insights and will consider this in future studies. This clarification will be added under Section 2.4.1 (Collection of Clinical Isolates).

3.3 Reviewer Comment (Page 12):

a) What is the relevance of bacteria characteristics in the materials and methods? Move this very relevant information to the discussion.

b) Where is the information on the biofilm formation?

Response:

a) We appreciate the reviewer’s suggestion and agree that the bacterial characteristics are more relevant to the Discussion section, under a new subheading titled "Relevance of Selected Bacterial Strains".

b) The information on biofilm formation is present under Section 2.4.5 (MBIC and MBEC assay), which includes details on how biofilms were formed and evaluated.

3.3 Reviewer Comment (Page 14):

a) Where is the information on the culturing conditions of Vero and its source?

b) Where is the description of the cytotoxicity assay?

Response:

a) The culturing conditions of Vero cells and their source is included in Subsection 2.5.1 and Subsection 2.5.2.

b) Cytotoxicity assay procedure were briefly described in Subsection 2.5.3 (Cytotoxicity assay of test samples) (Page 21). Detail experimental procedures were included in the supplementary information (Cytotoxicity Assay Procedure).

3.4 Reviewer Comment (Page 15): a) Statistical analysis comes usually at the end of the materials and methods section.

Response: Since different statistical parameter used for different assay, they are mentioned in the respective assay for relevance and clarity. For example, One-way ANOVA and Tukey's Post Hoc test procedure were outlined in the disk diffusion assay (Page 16).

3.5 Reviewer Comment (Page 21): a) Here, the authors have the cytotoxicity assay but they have mentioned it before. Please re-arrange the materials and methods section to be easier to interpret.

Response: Cytotoxicity Assay was divided into three subsections to ensure clarity. The title of subheading 2.4.2 (General preparation of media and controls for microbiological assays) was modified to avoid confusion with the Vero cell culture and Cytotoxicity Assay. Additionally, the cytotoxicity assay method was outlined before the microbiological assay.

4) Results

4.1 Reviewer Comment (Page 22): a) The reviewer does not understand why SPR is here, the assay is not mentioned in the materials and methods, concerning the chip, the solvents, the concentrations.

Response: The rationale of SPR will be clarified in the materials and methods- Section 2.3.1 (UV-Vis Spectroscopy)

4.2 Reviewer Comment:

(Page 23):

a) These NPs are very heterogeneous! Average size of 67? The results show at least three relevant peaks… What is the PDI? That value must be presented.

b) Why is the size increasing so much? 500 nm is huge! Do you know the in vivo consequences of that size?

(Page 24): a) Why is the size increasing so much?

Response:

(Page 23a): AgNPs was averaged 67.34 nm in diameter with polydispersity index (PDI) of 0.316 indicating moderate polydispersity (Section 3.1.1). In order to increase the homogeneity, the synthesized AgNPs were silica coated and amine functionalized, which resulted in unifo

---

## [Decision Letter · Decision Letter 2]

14 Aug 2025

Dear Dr. Ayubee,

Thank you for submitting your manuscript to PLOS ONE. After careful consideration, we feel that it has merit but does not fully meet PLOS ONE’s publication criteria as it currently stands. Therefore, we invite you to submit a revised version of the manuscript that addresses the points raised during the review process.

We look forward to receiving your revised manuscript.

Kind regards,

Thanh-Danh Nguyen, PhD

Academic Editor

PLOS ONE

Journal Requirements:

Reviewers' comments:

Reviewer's Responses to Questions

**Comments to the Author**

Reviewer #2: All comments have been addressed

Reviewer #4: (No Response)

2. Is the manuscript technically sound, and do the data support the conclusions?

Reviewer #2: Yes

Reviewer #4: Yes

3. Has the statistical analysis been performed appropriately and rigorously?

Reviewer #2: Yes

Reviewer #4: Yes

4. Have the authors made all data underlying the findings in their manuscript fully available?

Reviewer #2: Yes

Reviewer #4: Yes

5. Is the manuscript presented in an intelligible fashion and written in standard English?

Reviewer #2: Yes

Reviewer #4: Yes

Reviewer #2: The authors addressed all my comments; thus I think that the manuscript can be accepted for publication.

Reviewer #4: The manuscript has notably improved, and the authors have addressed most of my suggestions. However, there are several issues that must be corrected before publication:

The authors state:

“For the MBEC assay, biofilms were first allowed to form during a 24-hour incubation of 2 mL sterile MHB inoculated with 20 μL of a standardized bacterial suspension at 37°C. After washing with sterile deionized water to remove planktonic cells while keeping the biofilm attached to the sides of the test tube, 2 mL of dilutions of each test sample was added to the biofilm-containing test tubes. The tubes were incubated for another 4 hours at 37°C to evaluate the eradication of preformed biofilms under favorable bacterial growth conditions. The tubes were then washed with sterilized deionized water, stained with 0.1% crystal violet, and processed according to the protocol for the MBIC assay. The MBEC was determined to be the lowest concentration of the test sample showing an absorbance lower than 0.1, which is associated with total biofilm elimination.”

This definition is incorrect. What the authors are estimating with this protocol could be merely a biofilm disaggregation effect. Crystal violet staining cannot reveal bacterial viability within the biofilm. The correct definition is:

“The Minimum Biofilm Eradication Concentration (MBEC) is the lowest concentration of an antimicrobial agent that can kill all microorganisms within a biofilm.”

Please perform this experiment accordingly.

In Tables 6 and 7, what does “0” (zero) mean? As currently presented, it suggests that 0 μg/mL of AgNO₃ is capable of inhibiting biofilm development and eradicating all bacteria within the biofilm, which is not possible. This “0” value should be replaced with an exact number or, at the very least, indicated as “< [lowest tested value]”.

**Do you want your identity to be public for this peer review?** For information about this choice, including consent withdrawal, please see our Privacy Policy

Reviewer #2: No

Reviewer #4: No

---

## [Author Response · Author response to Decision Letter 3]

17 Aug 2025

Response to Reviewer #2:

We sincerely appreciate your kind feedback and are grateful for your positive evaluation of the revised manuscript. We are pleased to know that we have adequately addressed your comments. Thank you for your support.

Response to Reviewer #4:

We thank you for your thoughtful and constructive feedback, which has contributed significantly to improving the quality of our manuscript. We have carefully addressed all your concerns and made the necessary revisions.

1. Comment: Definition of the MBEC assay protocol

We appreciate your clarification regarding the MBEC assay and agree that the original definition did not accurately reflect the concept of biofilm eradication. In response to your comment, we have revised the manuscript to provide the correct definition. Specifically, we have updated the text to read:

The MBEC was determined to be the lowest concentration of the test sample that can kill all microorganisms within a biofilm, which is associated with biofilm disaggregation effect.

2. Comment: Clarification on Tables 6 and 7 – Meaning of “0”

We appreciate your observation regarding the “0” value in Tables 6 and 7, which actually represents that we didn’t find any growth inhibition with the tested concentrations of sample.

We acknowledge that this could be misleading and are grateful for your suggestion to clarify the data. In response, we have revised the tables and replaced the “0” value with “>100 [highest tested concentration]” to more accurately reflect that no tested concentration was effective and more concentrated sample is required for growth inhibition.

---

## [Editor Report · Decision Letter 3]

20 Aug 2025

Synergistic Antibacterial Action of AgNP-Ampicillin Conjugates: Evading β-Lactamase Degradation in Ampicillin-Resistant Clinical Isolates

PONE-D-24-53049R3

Dear Dr. Ayubee,

We’re pleased to inform you that your manuscript has been judged scientifically suitable for publication and will be formally accepted for publication once it meets all outstanding technical requirements.

Kind regards,

Thanh-Danh Nguyen, PhD

Academic Editor

PLOS ONE
---

## [Editor Report · Acceptance letter]

PONE-D-24-53049R3

PLOS ONE

Dear Dr. Ayubee,

I'm pleased to inform you that your manuscript has been deemed suitable for publication in PLOS ONE. Congratulations! Your manuscript is now being handed over to our production team.

Kind regards,

on behalf of

Dr. Thanh-Danh Nguyen

Academic Editor

PLOS ONE